# Functional hierarchy among different Rab27 effectors involved in secretory granule exocytosis

Kunli Zhao, Kohichi Matsunaga, Kouichi Mizuno, Hao Wang, Katsuhide Okunishi, Tetsuro Izumi*

Laboratory of Molecular Endocrinology and Metabolism, Department of Molecular Medicine, Institute for Molecular and Cellular Regulation, Gunma University, Maebashi, Japan

**Abstract** The Rab27 effectors are known to play versatile roles in regulated exocytosis. In pancreatic beta cells, exophilin-8 anchors granules in the peripheral actin cortex, whereas granuphilin and melanophilin mediate granule fusion with and without stable docking to the plasma membrane, respectively. However, it is unknown whether these coexisting effectors function in parallel or in sequence to support the whole insulin secretory process. Here, we investigate their functional relationships by comparing the exocytic phenotypes in mouse beta cells simultaneously lacking two effectors with those lacking just one of them. Analyses of prefusion profiles by total internal reflection fluorescence microscopy suggest that melanophilin exclusively functions downstream of exophilin-8 to mobilize granules for fusion from the actin network to the plasma membrane after stimulation. The two effectors are physically linked via the exocyst complex. Downregulation of the exocyst component affects granule exocytosis only in the presence of exophilin-8. The exocyst and exophilin-8 also promote fusion of granules residing beneath the plasma membrane prior to stimulation, although they differentially act on freely diffusible granules and those stably docked to the plasma membrane by granuphilin, respectively. This is the first study to diagram the multiple intracellular pathways of granule exocytosis and the functional hierarchy among different Rab27 effectors within the same cell.

## Editor's evaluation

This study examines how Rab27 and its different effectors regulate insulin secretory granule exocytosis. Using single and double knockouts and rescue experiments, the work presents convincing data to characterize the relative hierarchy of the Rab27 effectors, their action with the exocyst, and their roles in distinct types of exocytosis. Overall, this is a valuable study that sheds new light on the regulation of distinct forms of secretory granule exocytosis.

## Introduction

Synaptic vesicles must be docked and primed on the plasma membrane prior to stimulation for successful neuronal transmission to occur within 1 ms after electrical stimulation (*Südhof, 2013*). In contrast, secretory granules in exocrine and endocrine cells initiate exocytosis at a timepoint that is 1000-fold later than this, even when the $Ca^{2+}$ concentration is abruptly upregulated by caged-$Ca^{2+}$ compounds (*Kasai, 1999*). Their physiological release of bioactive substances likewise occurs much later. For example, digestive enzymes and insulin are secreted from pancreas cells continuously over a range of minutes to hours during food intake and subsequent hyperglycemia. To support such slow

*For correspondence:
tizumi@gunma-u.ac.jp

Competing interest: The authors declare that no competing interests exist.

and consecutive exocytosis, the exocytic pathway can be neither single nor linear, and the requisite steps for recruiting granules from the cell interior to the cell limits are therefore critical. In fact, total internal reflection fluorescence (TIRF) microscopy monitoring of prefusion behavior in living cells has revealed that insulin granules residing beneath the plasma membrane prior to stimulation, and those recruited from the cell interior after stimulation, fuse in parallel, with some variability in their ratios, during physiological glucose-stimulated insulin secretion (GSIS) (*Kasai et al., 2008*; *Ohara-Imaizumi et al., 2004*; *Shibasaki et al., 2007*). In many secretory cells, a greater number of granules are clustered in the actin cortex at the cell periphery and/or along the plasma membrane compared with other cytoplasmic areas. It has traditionally been considered that granules docked to the plasma membrane form a readily releasable pool, whereas those accumulated within the actin cortex form a reserve pool. However, this may be an oversimplification, considering that there is a gating system to prevent spontaneous or unlimited vesicle fusion in regulated exocytosis. In fact, multiple Rab27 effectors that are involved in intracellular granule trafficking show complex and differential effects on exocytosis (*Izumi, 2021*). For example, granuphilin (also known as exophilin-2 and Slp4) mediates stable granule docking to the plasma membrane but simultaneously prevents their spontaneous fusion by interacting with a fusion-incompetent, closed form of syntaxins (*Gomi et al., 2005*; *Torii et al., 2002*). Another effector, exophilin-8 (also known as MyRIP and Slac2-c), anchors secretory granules within the actin cortex (*Bierings et al., 2012*; *Desnos et al., 2003*; *Fan et al., 2017*; *Huet et al., 2012*; *Mizuno et al., 2011*; *Nightingale et al., 2009*), which is considered to have dual roles in accumulating granules at the cell periphery and in preventing their access to the plasma membrane. Although another effector, melanophilin (also known as exophilin-3 and Slac2-a), similarly captures melanosomes within the peripheral actin network in skin melanocytes (*Hammer and Sellers, 2012*), it mediates stimulus-induced granule mobilization and immediate fusion to the plasma membrane in pancreatic beta cells (*Wang et al., 2020*). It is unknown, however, just how the different Rab27 effectors coexisting in the same cell function in a coordinated manner to support the entire exocytic processes. Neither is it known whether multiple secretory pathways and/or rate-limiting steps exist in the final fusion stages of regulated granule exocytosis. To answer these questions, we must first seek to determine whether each effector functions in sequence or in parallel. In this study, we compared the exocytic profiles in beta cells lacking the two effectors with those in cells deficient in each single effector and in wild-type (WT) cells. This yielded valuable insights into the functional hierarchy and relationship among the exocytic steps in which individual effectors are involved. We also found that the exocyst, which universally functions in constitutive exocytosis (*Wu and Guo, 2015*), plays roles in the physical and functional connections between different Rab27 effectors.

## Results

### Melanophilin exclusively functions downstream of exophilin-8 to mediate the exocytosis of granules recruited from the actin cortex to the plasma membrane after stimulation

In monolayer mouse pancreatic beta cells, insulin granules were unevenly distributed with accumulation in the actin cortex (*Figure 1—figure supplement 1A*). Because the Rab27 effectors, melanophilin and exophilin-8, are known to show affinities to the actin motors, myosin-Va and/or -VIIa (*Hammer and Sellers, 2012*, *Izumi, 2021*), these effectors are presumed to function on granules within this peripheral actin network. In fact, they showed a similar uneven distribution and were colocalized especially at the cell periphery (*Figure 1—figure supplement 1B*). To examine the functional relationship between these effectors, we generated melanophilin/exophilin-8 double-knockout (ME8DKO) mice by crossing exophilin-8-knockout (Exo8KO) mice (*Fan et al., 2017*) with melanophilin-knockout (MlphKO) mice (see 'Materials and methods'). The doubly deficient beta cells displayed even distribution of insulin granules in the cytoplasm without accumulation at the cell periphery, in contrast to WT and MlphKO cells, but in a manner similar to that in Exo8KO cells (*Figure 1A*). Expression of exophilin-8, but not that of melanophilin, in ME8DKO cells at the endogenous level in WT cells redistributed them to the cell periphery (*Figure 1B*). This is consistent with the previous findings obtained from cells deficient in each single effector, which revealed that exophilin-8 is essential for granule accumulation within the actin cortex, whereas melanophilin is not (*Fan et al., 2017*; *Wang et al., 2020*).

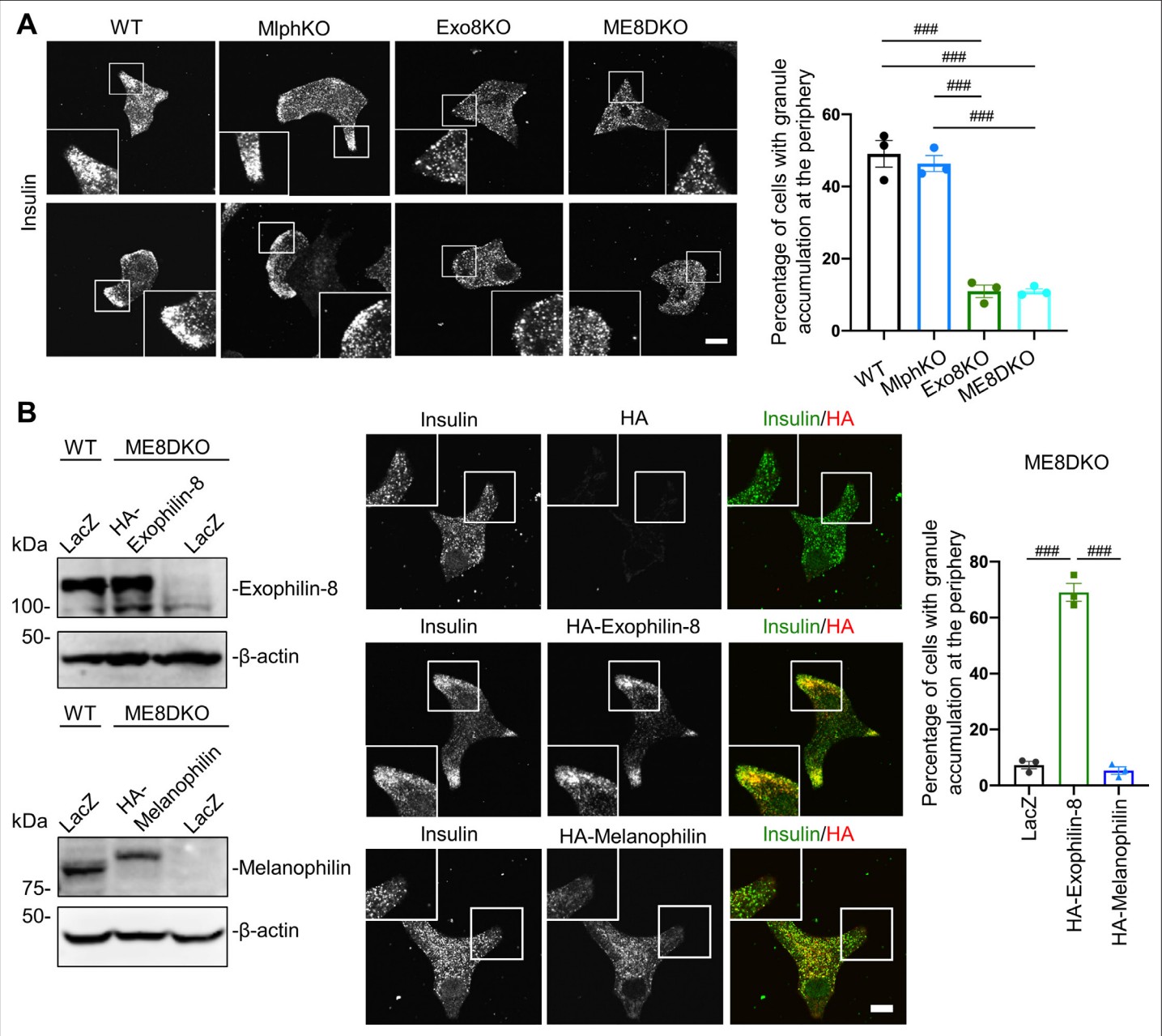

**Figure 1.** Exophilin-8, but not melanophilin, accumulates insulin granules in the action cortex. (**A**) WT, MlphKO, Exo8KO, and ME8DKO beta cells were immunostained with anti-insulin antibody. A peripheral accumulation of insulin immunosignals was quantified under confocal microscopy: clustering of insulin immunosignals at least in one corner (upper) or along the plasma membrane (lower) were counted as positive. More than 100 cells were inspected in each of three independent experiments. Note that Exo8KO and ME8DKO cells do not show the peripheral accumulation of insulin granules in contrast to WT and MlphKO cells. (**B**) ME8DKO cells were infected by adenovirus expressing either HA-exophilin-8 or HA-melanophilin at the endogenous protein levels found in WT cells. LacZ was expressed in WT and ME8DKO cells as controls. The cell extracts were immunoblotted with the indicated antibodies (left). The ME8DKO cells expressing LacZ (upper), HA-exophilin-8 (middle), and HA-melanophilin (lower) were immunostained with anti-insulin and anti-HA antibodies (center), and a peripheral accumulation of insulin was quantified as in (**A**) (right). Insets represent higher magnification photomicrographs of a cell within the region outlined by frames. Note that expression of HA-exophilin-8, but not HA-melanophilin, rescues the peripheral granule accumulation in ME8DKO cells. Bars, 10 μm. ###p<0.001 by one-way ANOVA.

The online version of this article includes the following source data and figure supplement(s) for figure 1:

**Source data 1.** Uncropped blot images of *Figure 1B*.

**Figure supplement 1.** Melanophilin and exophilin-8 are colocalized on insulin granules accumulated in the actin-rich cell periphery.

ME8DKO mice show glucose intolerance without changes in body weight or insulin sensitivity, as found in each singly deficient mice (*Figure 2—figure supplement 1A*). The cells deficient in melanophilin and/or exophilin-8 all showed decreases in GSIS compared with WT cells in both islet batch and perifusion assays, although the decreases in ME8DKO cells tended to be larger than those in MlphKO cells (*Figure 2A and B*, *Figure 2—figure supplement 1B and C*). We next monitored insulin granule exocytosis directly by TIRF microscopy in living cells expressing insulin fused with enhanced green fluorescent protein (EGFP). The numbers of visible granules under TIRF microscopy were not different among the four genotypes of cells (*Figure 2C*). Lack of the change in Exo8KO cells suggests that granules residing beneath the plasma membrane are not necessarily derived or supplied from granules anchored within the actin cortex. The amount of insulin released in the medium correlated well with a reduction in the total number of fusion events detected as a flash followed by diffusion of insulin-EGFP fluorescence during glucose stimulation in each of the mutant cells (*Figure 2D*, *Figure 2—figure supplement 1D*). We previously categorized fused insulin granules into three types depending on their prefusion behaviors (*Kasai et al., 2008*): those having been visible prior to stimulation, 'residents'; those becoming visible during stimulation, 'visitors'; and those invisible until fusion, 'passengers.' MlphKO cells with the genetic background of C57BL/6N mice showed a specific decrease in the passenger-type exocytosis, particularly in a later phase, compared with WT cells (*Figure 2D and E*), which is consistent with previous findings in parental *leaden* cells with the genetic background of C57BR/cdJ mice (*Wang et al., 2020*). In contrast, both Exo8KO and ME8DKO cells with the same C57BL/6N genetic background displayed decreases in both the resident and passenger types of exocytosis, while changes in the markedly less frequent visitor type were difficult to compare among the cells. Because there are no significant differences in exocytic phenotypes between Exo8KO and ME8DKO cells, melanophilin is thought to function downstream of exophilin-8. Considering the previously identified role of each effector in beta cells (*Fan et al., 2017*; *Wang et al., 2020*), the passenger-type exocytosis mediated by melanophilin via interactions with myosin-Va and syntaxin-4 appears to be derived from granules anchored in the actin cortex by exophilin-8.

## Exophilin-8 and melanophilin form a complex via the exocyst

To investigate the molecular basis by which exophilin-8 and melanophilin sequentially promote the passenger-type exocytosis, we first examined the expression level of each effector in cells lacking the other effector. Although the expression of exophilin-8 was not affected in MlphKO cells, that of melanophilin was decreased to 59.8 ± 9.0% (n = 4) in Exo8KO cells compared with WT cells (*Figure 3A*). However, this decrease should not affect GSIS in Exo8KO cells because a similar level of expression (65.4 ± 4.7%) in heterozygous MlphKO cells did not reduce it (*Figure 3—figure supplement 1*). We further found that the exophilin-8 and melanophilin expressed in the pancreatic beta cell line MIN6 form a complex (*Figure 3B*). These findings suggest that the protein stability of melanophilin partially depends on its interaction with exophilin-8, which is consistent with the model that melanophilin functions downstream of exophilin-8. However, they do not interact when expressed in HEK293A cells (*Figure 3—figure supplement 2A*), suggesting that they do not interact directly. The two effectors have been shown to interact with different proteins, except for Rab27a, in beta cells (*Fan et al., 2017*; *Wang et al., 2020*). The exophilin-8 mutant that loses its binding activity to RIM-BP2, and the melanophilin mutants that lose their binding activity to Rab27a, myosin-Va, or actin, all preserved their binding activity to the other WT effector in MIN6 cells (*Figure 3—figure supplement 2B*). To identify unknown intermediate proteins, we individually expressed Myc-TEV-FLAG (MEF)-tagged exophilin-8 and melanophilin in MIN6 cells, and we analyzed the proteins in each of the anti-FLAG immunoprecipitates using a liquid chromatography-tandem mass spectrometry (LC-MS/MS) (*Figure 3—figure supplement 3*). As a result, we identified the exocyst complex components, SEC8, SEC10, and EXO70, in both immunoprecipitates, which is consistent with the previous finding that exophilin-8 interacts with SEC6 and SEC8 in INS-1 832/13 cells (*Goehring et al., 2007*). We confirmed the endogenous interactions among melanophilin, exophilin-8, SEC6, and SEC10 in MIN6 cells (*Figure 3C*). The exocyst forms an evolutionarily conserved heterooctameric protein complex (*Wu and Guo, 2015*). To identify the components responsible for the interaction with two Rab27 effectors, we performed a visible immunoprecipitation (VIP) assay, which permits examination of large numbers of protein combinations and complicated one-to-many or many-to-many protein interactions simultaneously (*Katoh et al., 2015*). As a result, we found that exophilin-8 and melanophilin immediately, if not directly, interact

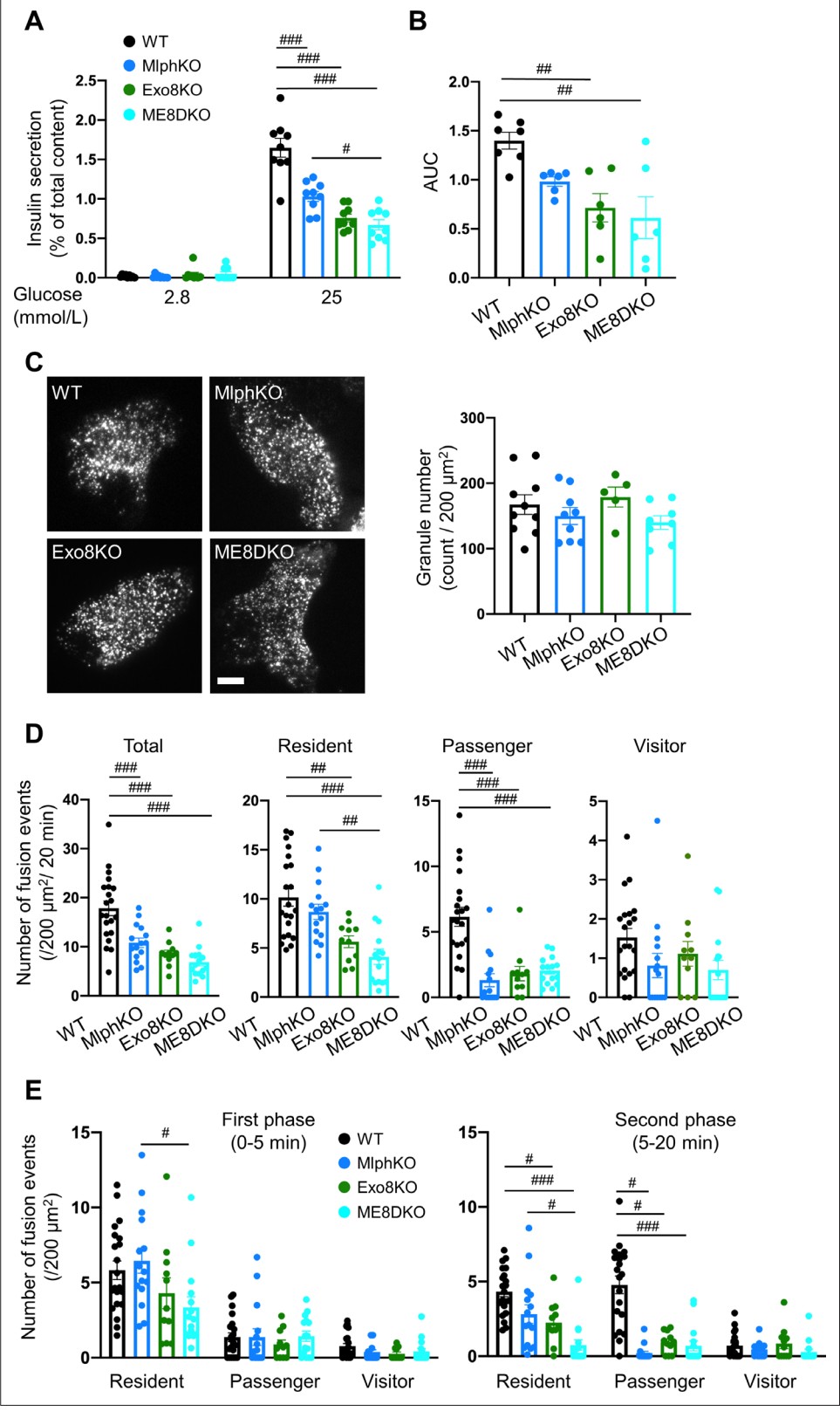

**Figure 2.** Insulin secretory defects of beta cells doubly deficient in melanophilin and exophilin-8 are indistinguishable from those singly deficient in exophilin-8. (**A**) Islets isolated from WT, MlphKO, Exo8KO, and ME8DKO mice at 12–18 weeks of age were preincubated in 2.8 mmol/L low glucose (LG)-containing KRB buffer at 37°C for 1 hr. They were then incubated in new LG buffer for 1 hr followed by 25 mmol/L high glucose (HG) buffer

*Figure 2 continued on next page*

*Figure 2 continued*

for 1 hr. The ratios of insulin secreted in the media to that left in the cell lysates (*Figure 2—figure supplement 1B*) are shown (left; n = 9 from three mice each). (**B**) Islets from age-matched mice (19- to 25-week-old) were perfused with 16.7 mmol/L glucose buffer for 30 min, and the ratios of insulin secreted in the media to that left in the cell lysates are plotted as in *Figure 2—figure supplement 1C*. The area under the curve (AUC) is shown (n = 9 from three mice each). (**C–E**) A monolayer of islet cells from the above four kinds of mice were infected with adenovirus encoding insulin-EGFP and were observed by total internal reflection fluorescence (TIRF) microscopy (**C**, left). The numbers of visible granules were manually counted for WT (n = 10 cells from three mice), MlphKO (n = 9 from three mice), Exo8KO (n = 5 cells from three mice), and ME8DKO (n = 8 cells from three mice) cells (**C**, right). Insulin granule fusion events in response to 25 mmol/L glucose for 20 min were counted as in *Figure 2—figure supplement 1D* for WT (n = 21 cells from five mice), MlphKO (n = 10 from three mice), Exo8KO (n = 12 cells from three mice), and ME8DKO (n = 12 cells from three mice) cells. The observed fusion events were categorized into three types: residents, visitors, and passengers (**D**). Summary of the three modes of fusion events (**E**) in the first phase (left, 0–5 min) and the second phase (right, 5–20 min). Note that the decrease in insulin exocytosis in ME8DKO cells is greater than that in MlphKO cells, specifically due to the decrease in the resident-type exocytosis. Bar, 10 µm. #p<0.05, ##p<0.01, ###p<0.001 by one-way ANOVA.

The online version of this article includes the following figure supplement(s) for figure 2:

**Figure supplement 1.** Phenotypic comparison among WT, MlphKO, Exo8KO, and ME8DKO mice.

---

with SEC8 and EXO70, respectively (*Figure 3D*). Although the exocyst components were found in both immunoprecipitates of melanophilin and exophilin-8, other interacting proteins such as RIM-BP2, RIM2, myosin-VIIa, and myosin-Va were identified in only one of the immunoprecipitates (*Figure 3E*), suggesting that the majority of each effector form a distinct complex in cells. In addition, although we previously reported that exophilin-8 primarily interacts with myosin-VIIa displaying a molecular mass of ~170 kDa in INS-1 832/13 rat cells (*Fan et al., 2017*), it interacted with mysosin-VIIa with an authentic mass of ~260 kDa in MIN6 mouse cells, which may reflect a species difference between these beta cell lines.

## The exocyst functions only in the presence of exophilin-8

In mammalian cells, the exocyst complex is assembled after separate formation of the subcomplex 1 (SEC3, SEC5, SEC6, SEC8) and the subcomplex 2 (SEC10, SEC15, EXO70, EXO84) (*Ahmed et al., 2018*). Because exophilin-8 and melanophilin immediately bind SEC8 and EXO70, respectively (*Figure 3D*), it is conceivable that the holo-exocyst links the two effectors. Previous VIP assays detected interactions between SEC8 or SEC3 in the subcomplex 1 and SEC10 in the subcomplex 2 (*Katoh et al., 2015*). Therefore, silencing of SEC10 is expected to efficiently disrupt the formation of the holo-exocyst. However, SEC10 knockdown by specific siRNAs did not affect the granule localization of exophilin-8 or melanophilin (*Figure 4—figure supplement 1*). Furthermore, there was no effect on the peripheral accumulation of insulin granules in contrast to the case in exophilin-8-deficient cells (*Figure 1A*). However, SEC10 knockdown significantly decreased the colocalization ratio of melanophilin, but not that of exophilin-8, with SEC6, another exocyst component that is colocalized with insulin granules in MIN6 cells (*Tsuboi et al., 2005*; *Figure 4A*). Considering that the two effectors specifically interact with the component of different subcomplexes (*Figure 3D*), it is reasonable that silencing of SEC10 that disrupts the subcomplex 2 dissociates melanophilin from exophilin-8 associated with the subcomplex 1. To obtain further evidence that the two effectors associate through the exocyst, we downregulated SEC8 that is thought to interact with exophilin-8 (*Figure 4—figure supplement 1*) and found that SEC8 knockdown markedly disrupted the complex formation (*Figure 4B*). These findings indicate that the exocyst physically connects the two effectors on the same granule.

SEC10 knockdown markedly decreased GSIS in WT cells, but induced no further decrease of GSIS in Exo8KO cells, which was already decreased by half compared with WT cells (*Figure 5A*, *Figure 5—figure supplement 1*). These findings suggest that the exocyst functions only in the presence of exophilin-8. As shown in *Figure 2C*, TIRF microscopy of Exo8KO cells expressing insulin-EGFP revealed no decrease in the number of visible granules (*Figure 5B*). Similarly, SEC10 knockdown does not affect the number of visible granules in either WT or Exo8KO cells. Consistent with insulin secretion assays (*Figure 5A*), it induces a marked decrease in glucose-stimulated fusion events in WT cells, but no additional decrease in Exo8KO cells (*Figure 5C*). Categorization by prefusion behavior revealed decreases in both resident and passenger types of exocytosis in WT cells, which phenocopied Exo8KO

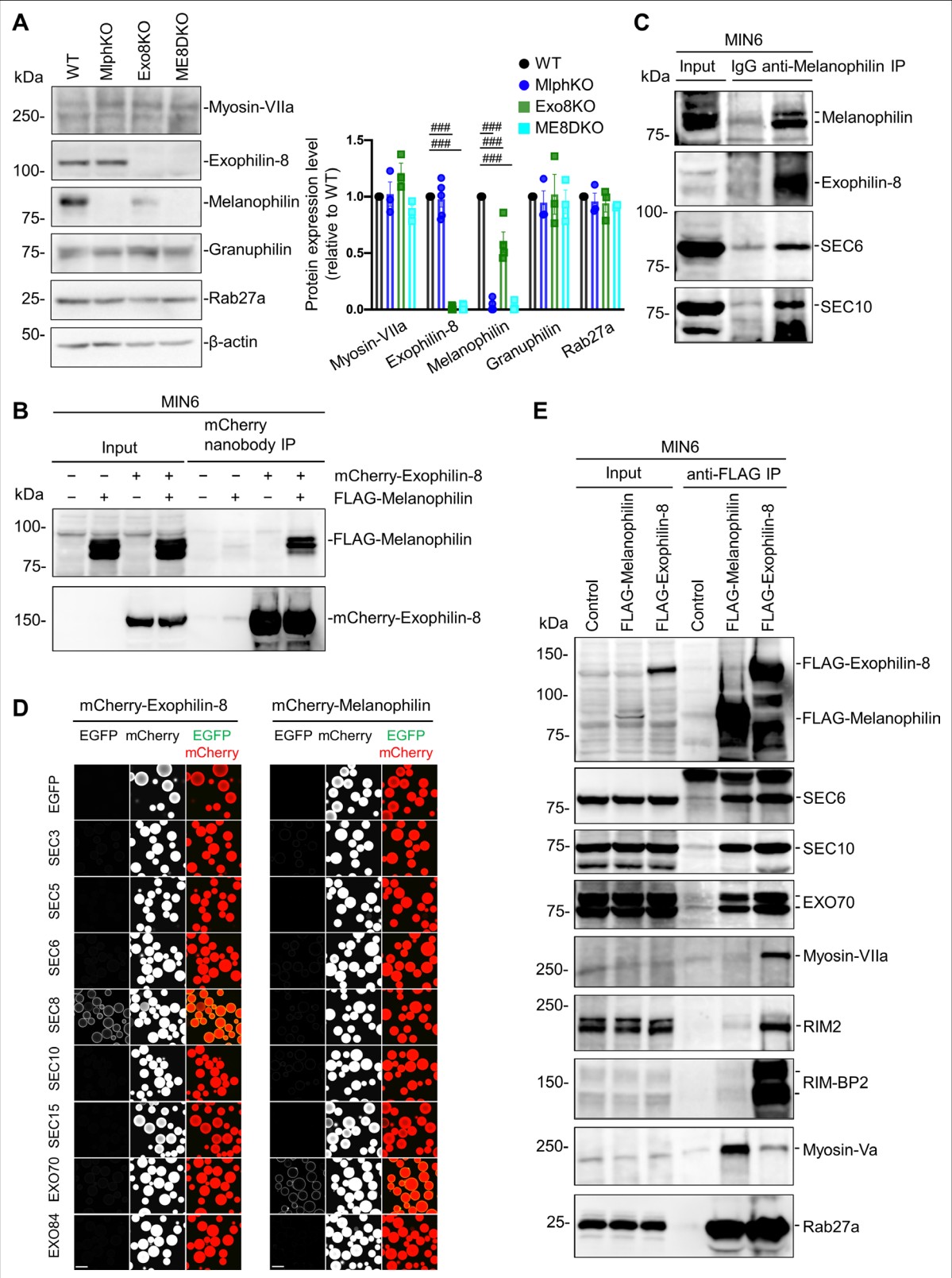

**Figure 3.** Exophilin-8 and melanophilin interact through the exocyst in beta cells. (**A**) Islet extracts (40 μg) from WT, MlphKO, Exo8KO, and ME8DKO mice were immunoblotted with antibodies toward the indicated proteins (left). Protein expression levels were quantified by densitometric analyses from 3 to 5 independent preparations (right). (**B**) MIN6 cells were infected with adenoviruses encoding control LacZ, mCherry-exophilin-8, and/or FLAG-melanophilin. After 2 days, the cell extracts underwent immunoprecipitation with mCherry nanobody. The immunoprecipitates (IP), as well as the 1%

*Figure 3 continued*

extracts (Input), were immunoblotted with anti-FLAG and anti-red fluorescent protein (RFP) antibodies. (**C**) MIN6 cell extracts were immunoprecipitated with rabbit anti-melanophilin antibody or control IgG, and the immunoprecipitates were immunoblotted with antibodies toward the indicated proteins. (**D**) HEK293A cells cultured in 10 cm dishes were transfected with mCherry-fused, exophilin-8 (left) or melanophilin (right) with the indicated EGFP-fused exocyst components. After 48 hr, the cell lysates were subjected to immunoprecipitation with mCherry-nanobody-bound glutathione-Sepharose beads. EGFP and mCherry fluorescence on the precipitated beads was observed by confocal microscopy. (**E**) MIN6 cells expressing FLAG-tagged, melanophilin or exophlin-8 were immunoprecipitated with anti-FLAG antibody, and the immunoprecipitates were immunoblotted with antibodies toward the indicated proteins. Note that the exocyst complex components exist in both immunoprecipitates, whereas RIM-BP2, RIM2, myosin-VIIa, and myosin-Va largely exist only in one of them. Bars, 100 µm. ###p<0.001 by one-way ANOVA.

The online version of this article includes the following source data and figure supplement(s) for figure 3:

**Source data 1.** Uncropped blot images of *Figure 3A, B, C and E*.

**Figure supplement 1.** Heterozygous MlphKO cells do not decrease insulin secretion despite the reduced melanophilin expression.

**Figure supplement 1—source data 1.** Uncropped blot images of *Figure 3—figure supplement 1A*.

**Figure supplement 2.** Exophilin-8 does not bind melanophilin directly or through the known interacting proteins.

**Figure supplement 2—source data 1.** Uncropped blot images of *Figure 3—figure supplement 2B*.

**Figure supplement 3.** Analysis of the melanophilin and exophilin-8 protein complexes.

cells without SEC10 knockdown (*Figure 2D*). Again, SEC10 knockdown had no effects on either type of exocytosis in Exo8KO cells. The holo-exocyst appears to function with exophilin-8 in the same pathway because SEC10 knockdown is expected to preserve the interaction between exophilin-8 and the subcomplex 1, as shown in *Figure 4A*. Taken together, it seems that exophilin-8 and the exocyst mediate the passenger-type exocytosis when they interact with melanophilin on the same granule.

## Exophilin-8 promotes the exocytosis of granules residing beneath the plasma membrane only in the presence of granuphilin

As shown in *Figure 2D*, exophilin-8 deficiency also affects the resident-type exocytosis. Another Rab27 effector, granuphilin, is thought to be deeply involved in this type of exocytosis because beta cells lacking granuphilin display very few granules directly attached to the plasma membrane under electron microscopy (*Gomi et al., 2005*). Despite this docking defect, these cells display a marked increase in granule exocytosis, possibly because granuphilin interacts with and stabilizes syntaxins in a fusion-incompetent, closed form (*Torii et al., 2002*). To explore the functional relationship between the two effectors in this type of exocytosis, we generated granuphilin/exophilin-8 double-knockout (GE8DKO) mice. The GE8DKO cells also exhibited an increase in GSIS compared with Exo8KO cells (*Figure 6A*, *Figure 6—figure supplement 1A*), which indicates that a significant number of granules fuse efficiently without exophilin-8 and granuphilin, thus without prior capture in the actin cortex and stable docking to the plasma membrane. TIRF microscopy of these cells expressing insulin-EGFP revealed that, although the number of visible granules was markedly decreased in granuphilin-knockout (GrphKO) cells compared with WT cells, as expected, it was not further decreased in GE8DKO cells compared with GrphKO cells (*Figure 6B*). The number of fusion events during glucose stimulation under TIRF microscopy was correlated with the amount of insulin released in the medium in each mutant cell (*Figure 6C*). All the types of exocytosis were increased in GrphKO cells, suggesting that the absence of granuphilin facilitates granule easier access to the fusion-competent machinery on the plasma membrane. Simultaneous absence of exophilin-8 induced strikingly differential effects on each type of exocytosis in GE8DKO cells: the increases in the passenger and visitor types were reduced to the level found in Exo8KO cells, whereas the increase in the resident type was completely unaffected. The former finding is consistent with the view that the passenger and visitor types of exocytosis are derived from granules captured in the actin cortex by exophilin-8. However, the latter finding indicates that, at least in the absence of granuphilin, exophilin-8 is dispensable for the exocytosis of granules already residing beneath the plasma membrane prior to stimulation.

However, the above findings were unexpected because exophilin-8 deficiency markedly affects the resident-type exocytosis in the presence of granuphilin (*Figure 2D*). We previously showed in WT cells that the resident-type exocytosis is heterogeneously derived from granuphilin-positive, immobile granules and granuphilin-negative, mobile granules (*Mizuno et al., 2016*). To directly assess the influence of exophilin-8 deficiency on granuphilin-mediated, docked granule exocytosis, we expressed

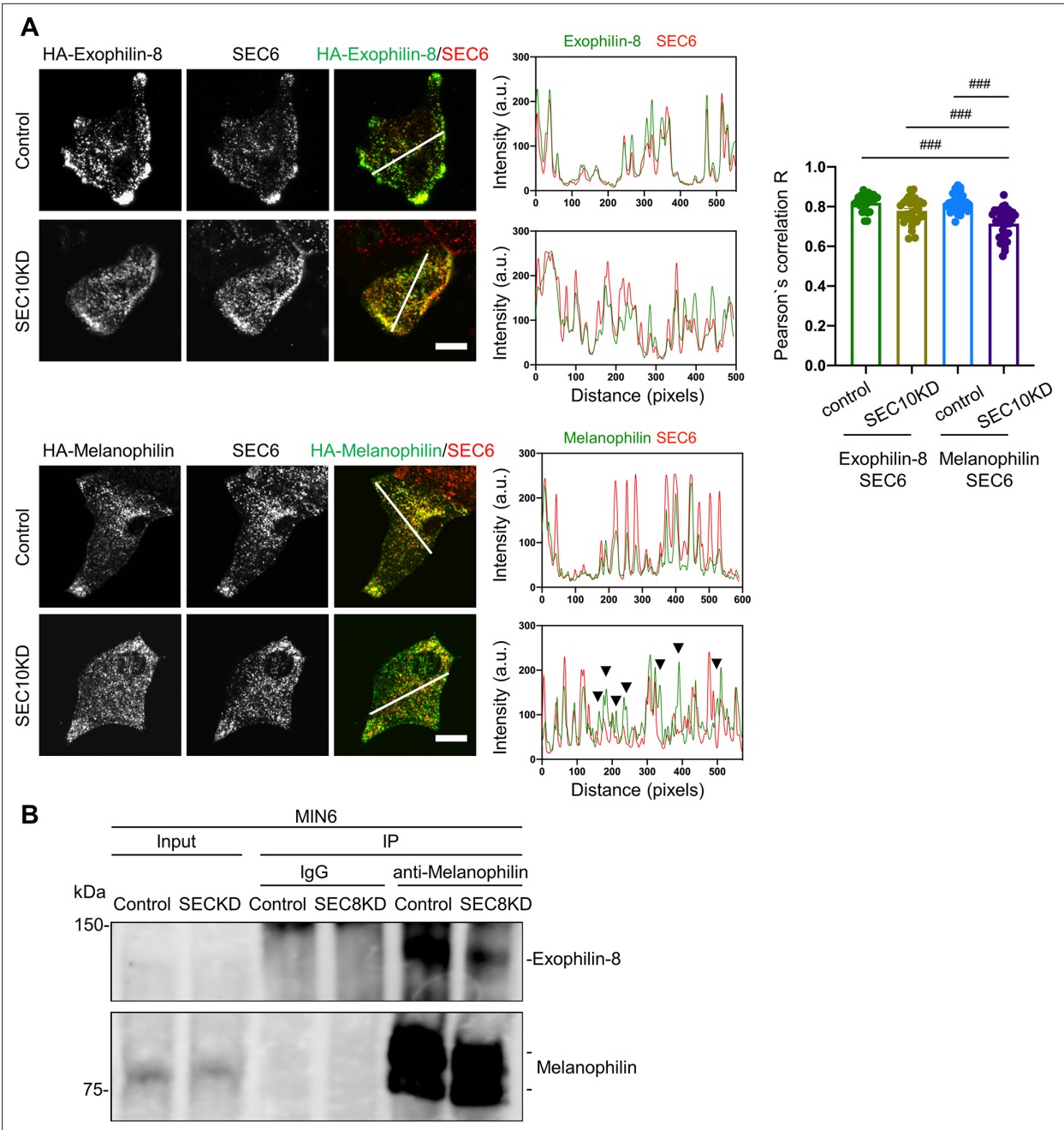

**Figure 4.** Knockdown of the exocyst component disrupts the interaction between melanophilin and exophilin-8. (**A**) HA-exophilin-8 and HA-melanophilin were expressed in Exo8KO and MlphKO monolayer beta cells, respectively, at the endogenous levels in WT cells as described in *Figure 1B*. They were then transfected with control siRNA or siRNA against SEC10 #11 or #12, as shown in *Figure 4—figure supplement 1A*. After fixation, the cells were coimmunostained with anti-HA and anti-SEC6 antibodies and were observed by confocal microscopy (left). Fluorescent intensity profiles along the indicated line of SEC6 and either HA-exophilin-8 or HA-melanophilin are shown (center). Colocalization was quantified by Pearson's correlation coefficient (right, n = 18–26 cells from two mice each). Note that SEC10 knockdown (KD) induces dissociation of melanophilin, but not exophilin-8, from SEC6 (black arrowheads). (**B**) MIN6 cells were transfected with control siRNA or siRNA against SEC8 #12, as shown in *Figure 4—figure supplement 1A*, and the cell extracts were subjected to immunoprecipitation with control IgG or anti-melanophilin antibody. The immunoprecipitates (IP), as well as the 1% extracts (Input), were immunoblotted with anti-exophilin-8 and anti-melanophilin antibodies. Bars, 10 µm. ###p<0.001 by one-way ANOVA.

The online version of this article includes the following source data and figure supplement(s) for figure 4:

**Source data 1.** Uncropped blot images of *Figure 4B*.

**Figure supplement 1.** Knockdown of the exocyst component does not affect the granule localization of exophilin-8 or melanophilin.

**Figure supplement 1—source data 1.** Uncropped blot images of *Figure 4—figure supplement 1A*.

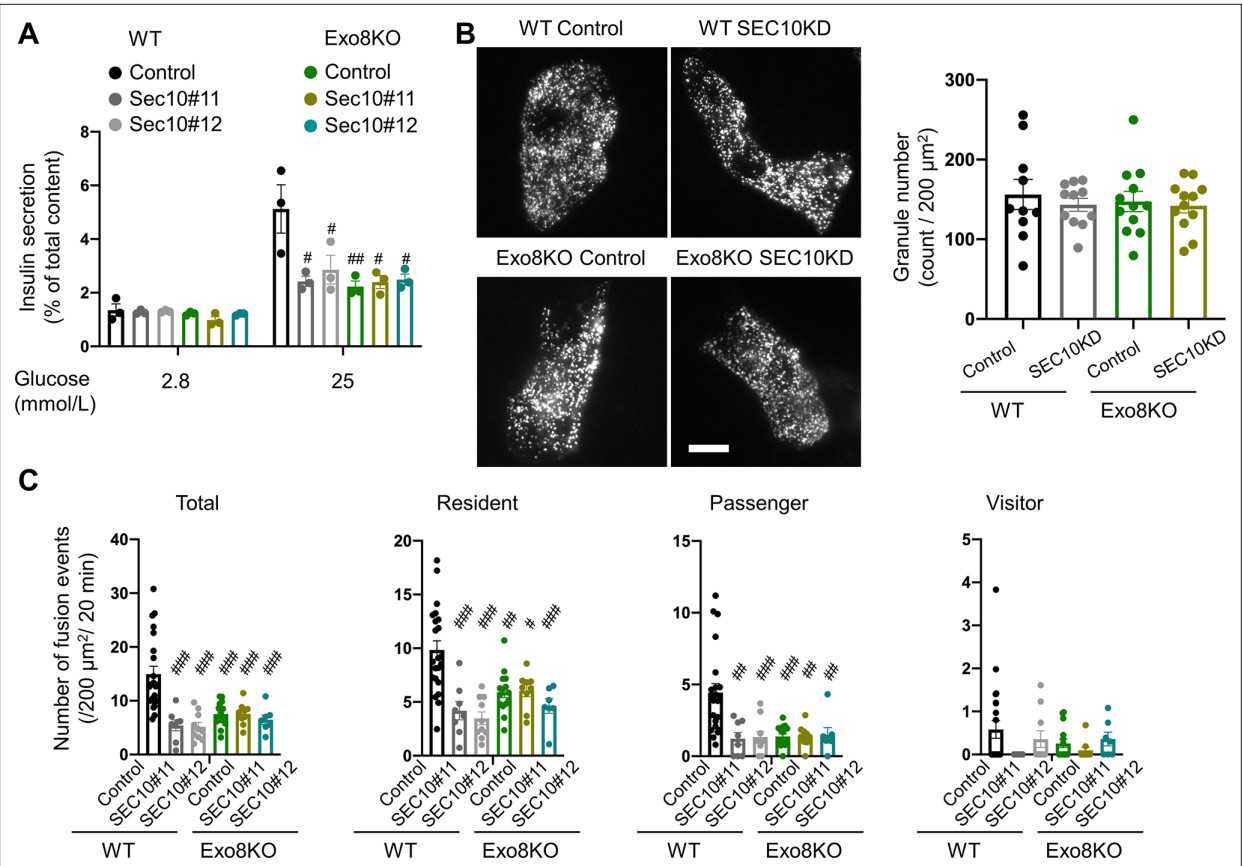

**Figure 5.** The exocyst affects insulin granule exocytosis only in the presence of exophilin-8. (**A–C**) WT or Exo8KO mouse islet cells were twice transfected with control siRNA or siRNA against SEC10 #11 or #12, as shown in *Figure 4—figure supplement 1A*, and were plated in a 24-well plate (**A**) or glass base dish (**B, C**). (**A**) The transfected monolayer cells were incubated for 1 hr in KRB buffer containing 2.8 mmol/L glucose and were then stimulated for 1 hr in the same buffer or in buffer containing 25 mmol/L glucose. Insulin levels secreted in the media and left in the cell lysates were measured (*Figure 5—figure supplement 1*), and their ratios are shown (n = 3 from three mice each). (**B**) The control or SEC10 knockdown (KD) cells (control siRNA-treated WT cells, n = 10 from three mice; SEC10 siRNA#11 or #12-treated WT cells, n = 11 from three mice; control siRNA-treated Exo8KO cells, n = 12 from three mice; SEC10 siRNA#11 or #12-treated Exo8KO cells, n = 12 from three mice) were infected with adenovirus encoding insulin-EGFP and were observed by total internal reflection fluorescence (TIRF) microscopy (left). The numbers of visible granules were manually counted (right). (**C**) The transfected cells (control siRNA-treated WT cells, n = 22 from three mice; SEC10 siRNA#11-treated WT cells, n = 8 from three mice; SEC10 siRNA#12-treated WT cells, n = 10 from three mice; control siRNA-treated Exo8KO cells, n = 15 from three mice; SEC10 siRNA#11-treated Exo8KO cells, n = 9 from three mice; SEC10 siRNA#12-treated Exo8KO cells, n = 7 from three mice) were infected with adenovirus encoding insulin-EGFP, and fusion events in response to 25 mmol/L glucose for 20 min were counted and categorized under TIRF microscopy as described in *Figure 2D*. Bar, 10 µm. #p<0.05, ##p<0.01, ###p<0.001 by one-way ANOVA versus control siRNA-treated WT cells.

The online version of this article includes the following figure supplement(s) for figure 5:

**Figure supplement 1.** Insulin levels left in the cell lysates in batch insulin secretion assays.

Kusabira Orange-1 (KuO)-fused granuphilin in GrphKO and GE8DKO cells (*Figure 6—figure supplement 1B*) because exogenous granuphilin expressed in the presence of endogenous granuphilin abnormally accumulates insulin granules beneath the plasma membrane and severely impairs their exocytosis (*Mizuno et al., 2016*; *Torii et al., 2004*; *Torii et al., 2002*). Under TIRF microscopy, the numbers of granuphilin-positive, -negative, and total granules were not significantly different between the two cell types. We confirmed similar numbers of these visible granules in WT and Exo8KO cells by immunostaining endogenous granuphilin and insulin (*Figure 6—figure supplement 1C*), suggesting that the expression levels of KuO-granuphilin in GrphKO and GE8DKO cells are properly adjusted to mimic WT and Exo8KO cells, respectively. Consistently, the number of total fusion events as well as the level of passenger-type exocytosis in GE8DKO cells expressing KuO-granuphilin (mimic Exo8KO cells) were markedly decreased compared with that in GrphKO cells expressing KuO-granuphilin (mimic WT cells; *Figure 6D*). Furthermore, although the levels of resident-type exocytosis from

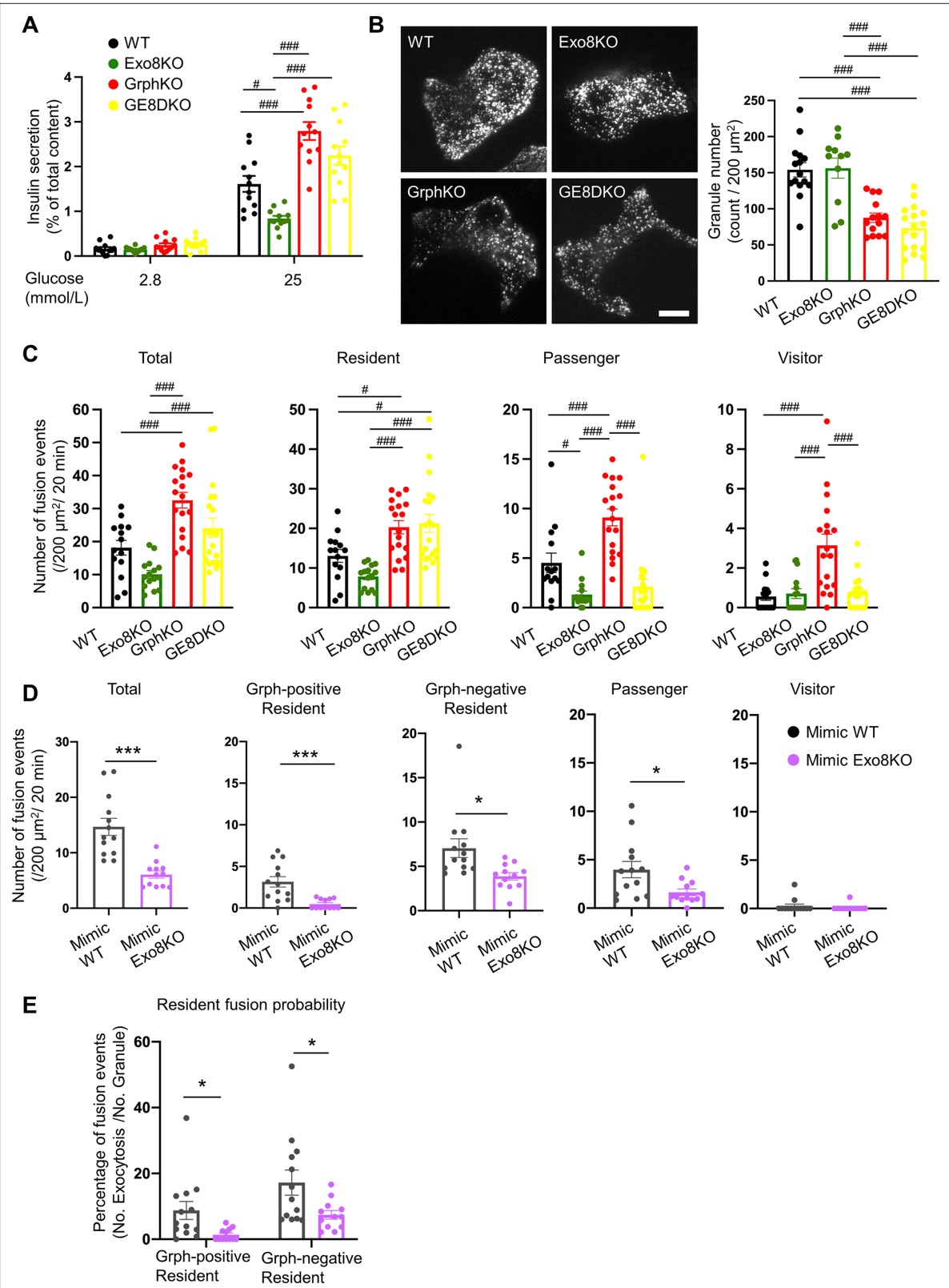

**Figure 6.** Exophilin-8 deficiency strongly inhibits the resident-type exocytosis from granuphilin-positive, stably docked granules. (**A**) Islets isolated from WT, Exo8KO, GrphKO, and GE8DKO mice at 12–17 weeks of ages were preincubated in low glucose (LG) KRB buffer for 1 hr. They were then incubated in another LG buffer for 30 min followed by high glucose (HG) buffer for 30 min. The ratios of insulin secreted in the media of that left in the cell lysates (*Figure 6—figure supplement 1A*) are shown as in *Figure 2A* (n = 12 from four mice each). (**B**) A monolayer of islet cells from the above four kinds

*Figure 6 continued on next page*

*Figure 6 continued*

of mice were infected with adenovirus encoding insulin-EGFP and were observed by total internal reflection fluorescence (TIRF) microscopy (left). The numbers of visible granules were manually counted for WT (n = 15 cells from four mice), Exo8KO (n = 11 cells from three mice), GrphKO (n = 14 cells from four mice), and GE8DKO (n = 17 cells from four mice) cells (right). (**C**) Fusion events in response to 25 mmol/L glucose for 20 min were counted and categorized as described in *Figure 2D* (WT: n = 14 cells from three mice; Exo8KO: n = 15 cells from three mice; GrphKO: n = 18 cells from three mice; GE8DKO: n = 20 cells from three mice). Note that the increases in the passenger and visitor types in GrphKO cells were eliminated by the simultaneous absence of exophilin-8 in GE8DKO cells, whereas the increase in the resident type was completely unaffected at all. (**D, E**) A monolayer of GrphKO (n = 13) and GE8DKO (n = 12) islet cells from three mice each were infected by adenoviruses encoding insulin-EGFP and KuO-granuphilin to mimic WT and Exo8KO cells, respectively, as described in *Figure 6—figure supplement 1B*. Insulin granule fusion events in response to 25 mmol/L glucose for 20 min were counted and categorized under TIRF microscopy as in *Figure 6C*, except that the granuphilin-positive and -negative granules were distinguished (**D**). There were no granuphilin-positive granules showing either passenger or visitor-type exocytosis. The fusion probability of granuphilin-positive and -negative granules is shown as the percentage of those granules displaying the resident-type exocytosis (**E**). Bar, 10 μm. #p<0.05, ###p<0.001 by one-way ANOVA. *p<0.05, *** p<0.001 by Student's *t* test.

The online version of this article includes the following figure supplement(s) for figure 6:

**Figure supplement 1.** Visible insulin granules associated with granuphilin under total internal reflection fluorescence (TIRF) microscopy.

both granuphilin-positive and -negative granules were significantly inhibited in mimic Exo8KO cells compared with mimic WT cells, the suppression was more complete toward granuphilin-positive granules. The fusion probability of granuphilin-positive granules, in which the number of fusion events is normalized by the number of visible granules, was also more strongly decreased (*Figure 6E*). These findings indicate that exophilin-8 primarily promotes the exocytosis of granules molecularly and stably tethered to the plasma membrane by granuphilin.

## The exocyst promotes the exocytosis of granules residing beneath the plasma membrane in the absence of granuphilin

Because the exocyst is also involved in the resident-type exocytosis (*Figure 5C*), we next investigated its functional relationship with granuphilin. We first examined the effects of SEC10 knockdown in GrphKO cells expressing insulin-EGFP under TIRF microscopy. Although it did not further decrease the number of visible granules in GrphKO cells (*Figure 7A*), as was the case in GE8DKO cells (*Figure 6B*), it markedly decreased all the types of exocytosis, including the resident-type exocytosis (*Figure 7B*), in contrast to the case in GE8DKO cells (*Figure 6C*). We then investigated the effect of SEC10 knockdown in GrphKO expressing both insulin-EGFP and KuO-granuphilin. Again, SEC10 knockdown did not change the numbers of granuphilin-positive, -negative, and total granules (*Figure 7—figure supplement 1*). However, these cells, including those having received control siRNA, did not respond to glucose stimulation well, possibly because two times of siRNA transfection and two kinds of adenovirus infection are too much intervention for vulnerable primary beta cells. Nevertheless, these cells did respond to KCl-induced depolarization. Although SEC10 knockdown significantly reduced the numbers of fusion from both granuphilin-positive and -negative granules (*Figure 7C*), it specifically decreased the fusion probability of granuphilin-negative granules (*Figure 7D*). Therefore, the exocyst deficiency primarily affects the resident-type exocytosis from granuphilin-negative granules, in contrast to exophilin-8 deficiency (*Figure 6E*).

## Discussion

In this study, using distinct prefusion behaviors observed by TIRF microscopy as markers of differential exocytic routes, we investigated the functional hierarchy among different Rab27 effectors in pancreatic beta cells lacking one or two of these effectors. We first present evidence that exophilin-8 functions upstream of melanophilin to drive the passenger-type exocytosis. This type of exocytosis appears to be derived from granules within the actin network because exophilin-8 is essential for granule accumulation in the actin cortex and both effectors function via interactions with myosin-VIIa and myosin-Va, respectively, in beta cells (*Fan et al., 2017*; *Wang et al., 2020*). We also show that these two effectors are linked by the exocyst protein complex. Although this evolutionarily conserved protein complex is known to function in constitutive exocytosis, it has not been established whether it plays a universal role in regulated exocytosis. For example, *Drosophila* with mutation in SEC5 displays a defect in neurite outgrowth but not in neurotransmitter secretion (*Murthy et al., 2003*). In yeast, the

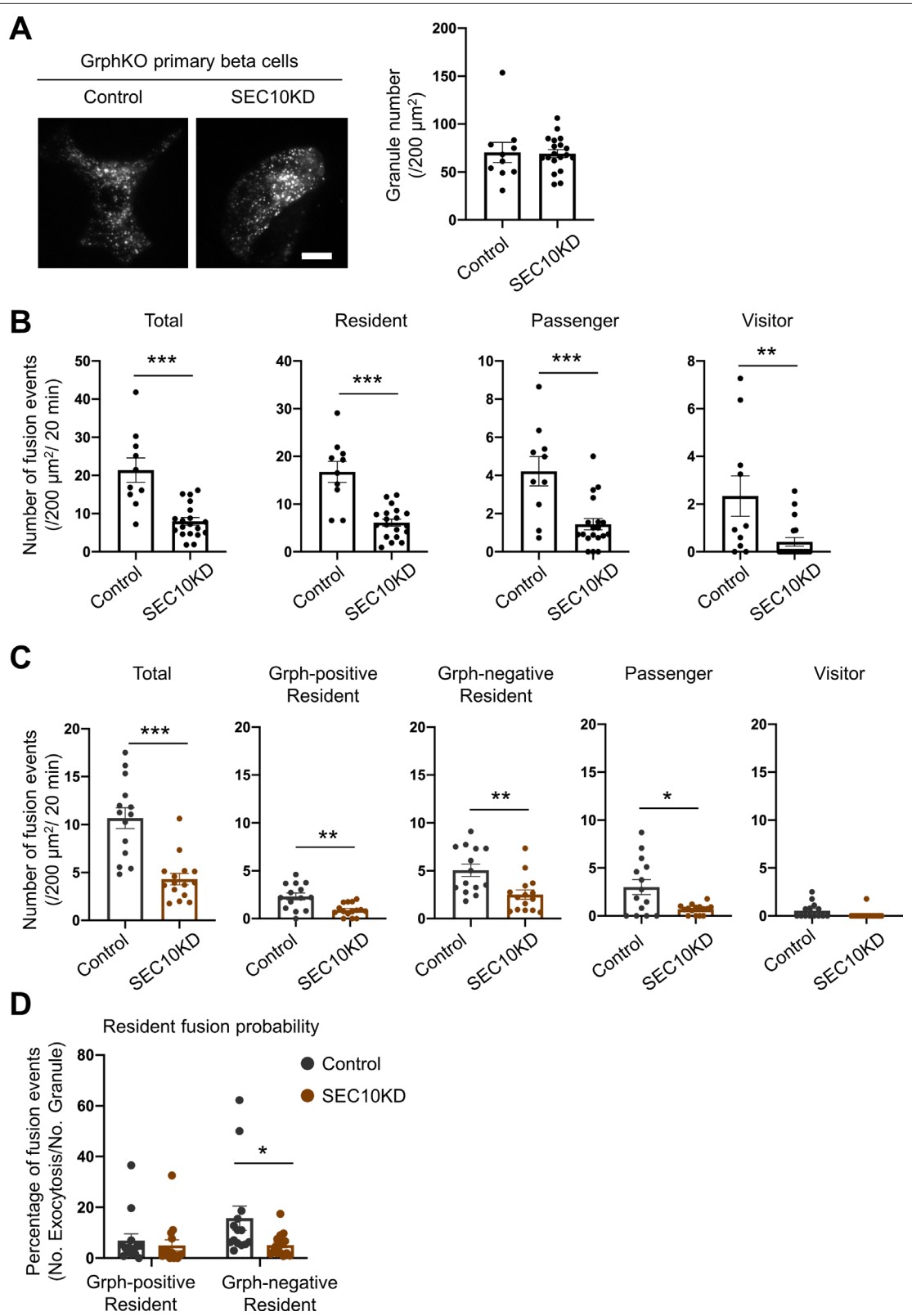

**Figure 7.** Exocyst deficiency strongly inhibits the resident-type exocytosis from granuphilin-negative, untethered granules. (**A**) GrphKO beta cells were transfected with control siRNA (n = 10 from three mice) or siRNA against SEC10 #11 or #12 (n = 19 from three mice) were infected with adenovirus encoding insulin-EGFP as described in *Figure 5B*. They were observed by total internal reflection fluorescence (TIRF) microscopy (left), and numbers of visible granules were manually counted (right). (**B**) The control (n = 10 from three mice) or SEC10 knockdown (KD) cells (n = 19 from three mice) were

*Figure 7 continued on next page*

*Figure 7 continued*

infected with adenovirus encoding insulin-EGFP, and fusion events in response to 25 mmol/L glucose for 20 min were counted and categorized under TIRF microscopy as described in *Figure 6C*. Note that in contrast to exophilin-8 knockout (*Figure 6C*), SEC10 knockdown suppresses the resident-type exocytosis in GrphKO cells. (**C, D**) GrphKO islet cells from three mice were transfected with control (n = 14) or SEC10 siRNAs (n = 15) twice, and were infected with adenoviruses encoding insulin-EGFP and KuO-granuphilin on the next day. Because these cells that had undergone two times siRNA transfection and two kinds of adenovirus infection failed to respond to 25 mmol/L glucose stimulation well, they were depolarized by 60 mmol/L potassium. Fusion events during 6 min were counted and categorized (**C**) as in *Figure 6D*. The fusion probability of granuphilin-positive and -negative granules is shown (**D**) as in *Figure 6E*. Note that SEC10 knockdown selectively suppresses it from granuphilin-negative granules. Bar, 10 µm. *p<0.05, **p<0.01, ***p<0.001 by Student *t* test.

The online version of this article includes the following figure supplement(s) for figure 7:

**Figure supplement 1.** Visible insulin granules associated with granuphilin under total internal reflection fluorescence (TIRF) microscopy.

exocyst interacts with Sec4, a member of the Rab protein on secretory vesicles, via exocyst component Sec15 (*Guo et al., 1999*). Further, Sec4 and Sec15 directly bind Myo2, the yeast myosin-V, for secretory vesicle transport (*Jin et al., 2011*). The exocyst components also interact with the SNARE fusion machinery: Sec6 with Snc2 (VAMP/synaptobrevin family in mammals) (*Shen et al., 2013*) and Sec9 (SNAP-25 family in mammals) (*Dubuke et al., 2015*), and Sec3 with Sso2 (syntaxin family in mammals) (*Yue et al., 2017*). Although proteins corresponding to Rab27 effectors are absent in yeast, they appear to be involved in similar interactions in mammalian cells. In fact, melanophilin interacts with Rab27a, myosin-Va, and syntaxin-4 in beta cells (*Wang et al., 2020*). In mammalian cells, the exocyst complex is assembled after the formation of two separate subcomplexes (*Ahmed et al., 2018*). The holo-exocyst appears to connect exophilin-8 and melanophilin because both subcomplex components commonly exist in the immunoprecipitates of two effectors in beta cells. We further show that exophilin-8 and melanophilin associate via different subcomplex components, SEC8 and EXO70, respectively, and that disruption of one subcomplex results in dissociation between the two effectors. To our knowledge, this is the first example showing that different effectors toward the same Rab form a complex in cells, which corroborates the previous suggestion that multiple effectors on the same granules may smooth the transition between consecutive intermediate exocytic processes (*Izumi, 2021*).

We then show that exophilin-8 knockout and SEC10 knockdown exhibit very similar defects in granule exocytosis in WT cells, indicating that they function together. Furthermore, neither affects the number of granules visualized by TIRF microscopy. The sole difference is that, although exophilin-8 deficiency erases the granule accumulation in the actin cortex, exocyst deficiency does not, which suggests that the exocyst functions downstream of exophilin-8. Consistent with this view, we could not find any effects of SEC10 knockdown in Exo8KO cells. In the pheochromocytoma cell line, PC12, the nascent granules generated are transported in a microtubule-dependent manner to the cell periphery within a few seconds (*Rudolf et al., 2001*). In skin melanocytes, melanosomes are dispersed throughout the cytoplasm using the myosin-Va motor along dynamic actin tracks assembled by the SPIRE actin nucleator (*Alzahofi et al., 2020*). Thus, even without prior capture in the actin cortex by exophilin-8, insulin granules may also be eventually transported close to the plasma membrane using these routes.

We next show that, although exophilin-8 deficiency inhibits the resident-type exocytosis in WT cells, it does not affect it despite its increase in GrphKO cells. However, in GrphKO cells expressing KuO-granuphilin, exophilin-8 deficiency almost completely inhibits the exocytosis of granuphilin-positive granules. Granuphilin-mediated, stably docked granules are thought to require priming machinery for fusion, such as Munc13, that converts a granuphilin-associated, closed form of syntaxin to a fusion-competent, open form (*Mizuno and Izumi, 2022*). Exophilin-8 can contribute to this process because it is associated with RIM-BP2, RIM, and Munc13 (*Fan et al., 2017*), which are known to have such a priming role in synaptic vesicle exocytosis (*Brockmann et al., 2019*; *Brockmann et al., 2020*). These priming factors could nonspecifically convert even granuphilin-free syntaxins into the open form because exophilin-8 deficiency also significantly inhibits the resident-type exocytosis from granuphilin-negative granules. In fact, the exophilin-8 mutant that loses binding activity to RIM-BP2 has no effect on the decreased insulin secretion in exophilin-8-deficient cells (*Fan et al., 2017*).

In contrast to exophilin-8 deficiency, SEC10 deficiency suppresses the resident-type exocytosis even in the absence of granuphilin. Furthermore, in GrphKO cells expressing KuO-granuphilin, it

specifically decreases the fusion probability from granuphilin-negative granules showing this type of exocytosis. We previously showed that granuphilin-negative granules are more mobile and fusogenic compared with granuphilin-positive granules under TIRF microscopy (*Mizuno et al., 2016*). However, these diffusible granules must somehow be directionally mobilized to the plasma membrane for fusion. The exocyst likely functions in this process, considering that it generally tethers secretory vesicles to the plasma membrane prior to membrane fusion in a broad range of cells (*Wu and Guo, 2015*). This view also explains why SEC10 deficiency does not affect the fusion probability of granuphilin-positive granules that have already stably docked to the plasma membrane. However, this tethering step should occur within 100–200 nm distance from the plasma membrane because SEC10 knockdown has no effects on the numbers of granules visualized by TIRF microscopy whether granuphilin is present or not.

In summary, our findings are the first to determine the functional hierarchy among different Rab27 effectors expressed in the same cell, although the exact mechanism remains unknown how these

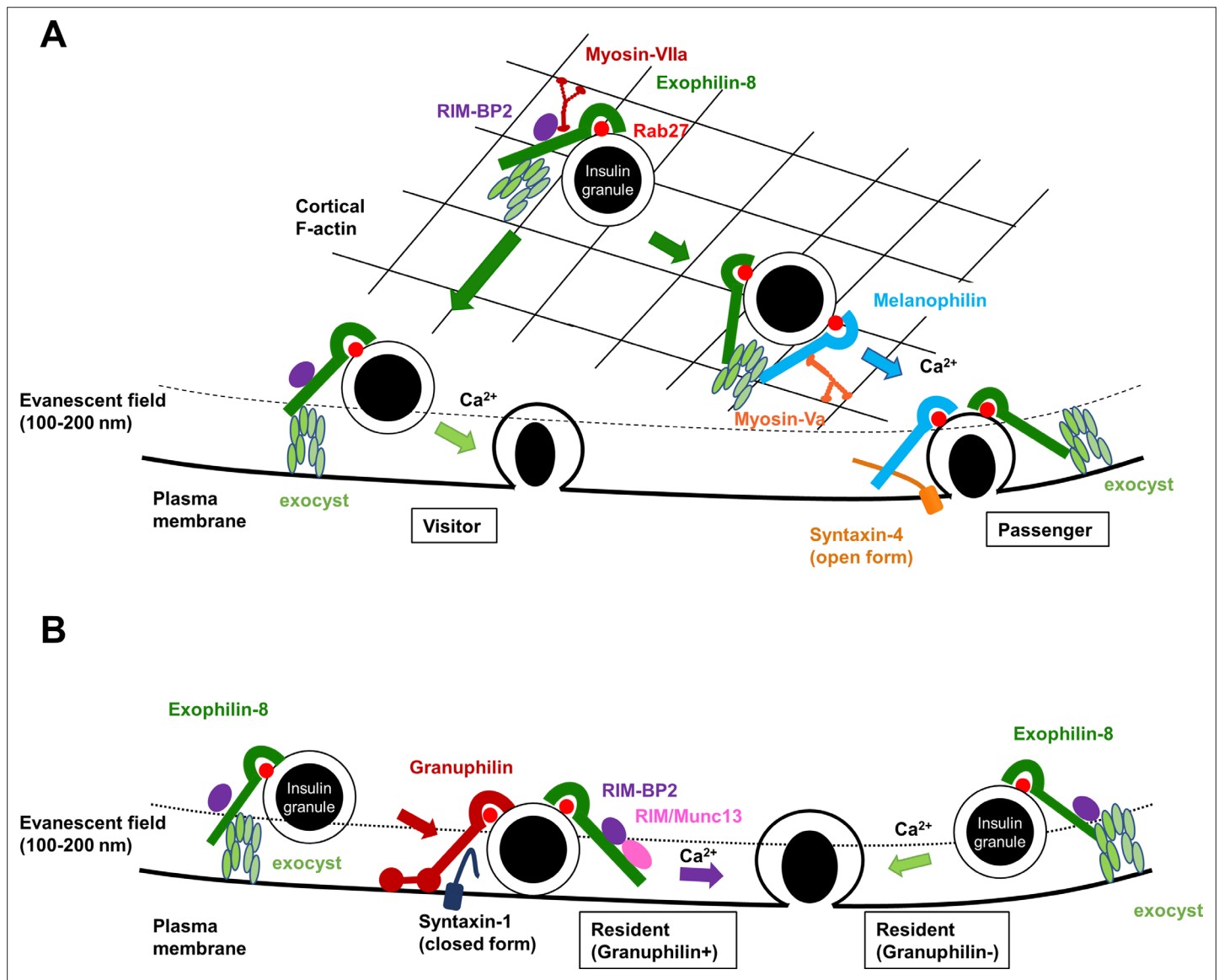

**Figure 8.** A schematic model for the functional relationship among different Rab27 effectors and the exocyst in insulin granule exocytosis. At least, four different types of insulin granule exocytosis are discriminated by total internal reflection fluorescence (TIRF) microscopic analyses: the passenger type and the visitor type derived from granules anchored in the actin cortex (**A**), and the resident type from granules docked to the plasma membrane or from granules that remain untethered beneath the plasma membrane (**B**). Exophilin-8, melanophilin, granuphilin, and the exocyst differentially regulate each type of exocytosis. See details in 'Discussion.'

different steps are regulated in a coordinated manner. Some granules trapped in the actin cortex by the action of exophilin-8 stimulus-dependently fuse either immediately by the action of melanophilin (the passenger-type exocytosis) or after staying beneath the plasma membrane for a while (the visitor-type exocytosis) (*Figure 8A*). The exocyst may help tether granules to the plasma membrane in these types of exocytosis. Other granules somehow transported to the cell periphery without capture in the actin cortex are stably tethered to the plasma membrane by the action of granuphilin or remain untethered beneath the plasma membrane (*Figure 8B*). The former granules stimulus-dependently fuse after the granuphilin-associated, closed form of syntaxins are converted into the fusion-competent, open form, possibly by the action of exophilin-8-associated priming factors, such as RIM-BP2. The latter granules fuse stochastically after directed to the plasma membrane by the action of the exocyst. It should be noted, however, that granule exocytosis is not abrogated in beta cells lacking any of the single or double Rab27 effectors. This may suggest the existence of other secretory pathways. For example, other Rab27 effectors, such as exophilin-7 (also known as JFC1 or Slp1), can also promote some types of exocytosis (*Wang et al., 2013*). However, given that granules can arrive close to the plasma membrane and fuse efficiently without prior capture in the actin cortex by exophilin-8 and/or stable docking to the plasma membrane by granuphilin, Rab27 effectors are thought to be evolved to play regulatory roles that prevent spontaneous or inappropriate fusion in regulated secretory pathways, as suggested previously (*Izumi, 2021*). In any case, it is now evident that there are multiple redundant paths and rate-limiting processes toward the final fusion step in granule exocytosis, which evolved to ensure the robust performance of the mechanisms governing the secretion of vital molecules, such as insulin.

# Materials and methods

## Key resources table

| Reagent type (species) or resource | Designation | Source or reference | Identifiers | Additional information |
|---|---|---|---|---|
| Gene (*Mus musculus*) | MyRIP (Exophilin-8) | GenBank | Gene ID: 245049 | |
| Gene (*M. musculus*) | Mlph (Melanophilin) | GenBank | Gene ID: 171531 | |
| Gene (*M. musculus*) | Sytl4 (Granuphilin) | GenBank | Gene ID: 27359 | |
| Cell line (*Homo sapiens*) | HEK293A | Invitrogen | Cat#: R70507; RRID: CVCL_6910 | The cell line has been authenticated and tested negative for mycoplasma |
| Cell line (*M. musculus*) | Insulinoma | *Miyazaki et al., 1990* | RRID:CVCL_0431 | The cell line has been authenticated and tested negative for mycoplasma |
| Transfected construct (*M. musculus*) | On-Target plus non-targeting pool siRNA | Horizon Discovery Ltd | Cat#: D-001810-10-05 | Silencer Select |
| Transfected construct (*M. musculus*) | siRNA to Exoc5 (SEC10) | Horizon Discovery Ltd | Cat#: 105504 | On-Target plus Set of 4 siRNA (J-047583-11 and -12) |
| Transfected construct (*M. musculus*) | siRNA to Exoc4 (SEC8) | Horizon Discovery Ltd | Cat#: 20336 | On-Target plus Set of 4 siRNA (J-051541-12-0050) |
| Biological sample (*M. musculus*) | Primary pancreatic beta cells | This paper | | Freshly isolated islets from male mouse pancreas |
| Antibody | Anti-Rab27a/b (rabbit polyclonal) | IBL | Cat#: 18975; RRID:AB_494635 | WB (1:2000) |
| Antibody | Anti-Rab27a (mouse monoclonal) | BD Biosciences | Cat#: 558532; RRID: AB_647327 | IF (1:100) |
| Antibody | Anti-Exophilin-8 (goat polyclonal) | Abcam | Cat#: ab10149; RRID:AB_296882 | IF (1:100) |
| Antibody | Anti-Melanophilin (goat polyclonal) | Everest Biotech | Cat#: EB05444; RRID:AB_2146092 | WB (1:2000) |

*Continued on next page*

*Continued*

| Reagent type (species) or resource | Designation | Source or reference | Identifiers | Additional information |
|---|---|---|---|---|
| Antibody | Anti-Melanophilin (rabbit polyclonal) | Proteintech | Cat#: 10338-1-AP; RRID: AB_2146104 | WB (1:2000) |
| Antibody | Anti-Myosin-VIIa (rabbit polyclonal) | Abcam | Cat#: ab3481; RRID: AB_303841 | WB (1:2000) |
| Antibody | Anti-Myosin-Va (rabbit polyclonal) | Cell Signaling Technology | Cat#: 3402; RRID: AB_2148475 | WB (1:2000) |
| Antibody | Anti-RIM-BP2 (rabbit polyclonal) | Proteintech | Cat#: 15716-1-AP; RRID:AB_2878173 | WB (1:2000) |
| Antibody | Anti-RIM1/2 (rabbit polyclonal) | Synaptic Systems | Cat#: 140213; RRID: AB_2832237 | WB (1:2000) |
| Antibody | Anti-SEC6 (mouse monoclonal) | Assay Designs | Cat#: ADI-VAM-SV021; RRID: AB_10618264 | IF (1:100), WB (1:2000) |
| Antibody | Anti-SEC8 (rabbit polyclonal) | Proteintech | Cat#: 11913-1-AP; RRID: AB_2101565 | WB (1:2000) |
| Antibody | Anti-SEC10 (rabbit polyclonal) | Proteintech | Cat#: 17593-1-AP; RRID: AB_2101582 | WB (1:2000) |
| Antibody | Anti-EXO70 (rabbit polyclonal) | Proteintech | Cat#: 12014-1-AP; RRID: AB_2101698 | WB (1:2000) |
| Antibody | Anti-FLAG (rabbit polyclonal) | Sigma-Aldrich | Cat#: F7425; RRID: AB_439687 | WB (1:2000) |
| Antibody | Anti-HA (rabbit polyclonal) | MBL | Cat#: 561; RRID: AB_591839 | WB (1:2000) |
| Antibody | Anti-HA (rat monoclonal) | Roche | Cat#: 11867423001; RRID: AB_390918 | IF (1:100) |
| Antibody | Anti-β-actin (mouse monoclonal) | Sigma-Aldrich | Cat#: A5316; RRID: AB_476743 | WB (1:10,000) |
| Antibody | Anti-red fluorescent protein (RFP) (rabbit polyclonal) | MBL | Cat#: PM005; RRID:AB_591279 | WB (1:10,000) |
| Commercial assay or kit | AlphaLISA insulin kit | PerkinElmer | Cat#: AL350HV/C/F | Insulin detection |
| Commercial assay or kit | Insulin high range kit | Cisbio | Cat#: 62IN1PEG | Insulin detection |
| Commercial assay or kit | Insulin ultra sensitive kit | Cisbio | Cat#: 62IN2PEH | Insulin detection |
| Recombinant DNA reagent | pEGFP-C3 (plasmid) | Addgene | Cat#: 53755-53762; RRID: Addgene_53755–53762 | GFP version of SEC3, SEC5, SEC6, SEC8, SEC10, SEC15, EXO70, and EXO84 |
| Software, algorithm | ImageQuant TL software | Cytiva | RRID:SCR_014246 | Quantify immunoblotting |
| Software, algorithm | NIS Element Viewer | Nikon | RRID:SCR_014329 | Quantify colocalization and analyze fusion events |

*Continued on next page*

*Continued*

| Reagent type (species) or resource | Designation | Source or reference | Identifiers | Additional information |
|---|---|---|---|---|
| Software, algorithm | Protein pilot software | SCIEX | RRID:SCR_018681 | Mass spectrometry |
| Other | Rhodamine-conjugated phalloidin | Thermo Fisher Scientific | Cat#: R415 | IF (1:100) |
| Other | Anti-FLAG M2 Affinity Gel | Sigma-Aldrich | Cat#: A2220 | IP |
| Other | Protein G Sepharose 4 Fast Flow | GE Healthcare Biosciences | Cat#: GE17-0618-01 | IP |
| Other | Lipofectamine 3000 | Invitrogen | Cat#: L3000001 | Transfection |
| Other | Lipofectamine RNAiMax | Invitrogen | Cat#: 13778075 | Transfection |

## Mice and pancreatic islet cell preparation

Animal experiments were performed according to the rules and regulations of the Animal Care and Experimental Committees of Gunma University (permit number: 22-010; Maebashi, Japan). Only male mice and their tissues and cells were phenotypically characterized in this study. *Leaden* (C57J/L) mice with nonfunctional mutation of the gene encoding melanophilin (*Mlph*) (*Matesic et al., 2001*) were purchased from The Jackson Laboratory (Strain #:000668, RRID:IMSR_JAX:000668), and were back-crossed with C57BL/6N mice 10 times to generate MlphKO mice. Exo8KO mice in the genetic background of C57BL/6N mice were described previously (*Fan et al., 2017*). ME8DKO mice were obtained by mating Exo8KO mice with the MlphKO mice described above. GrphKO mice in the genetic background of C3H/He mice were described previously (*Gomi et al., 2005*). The male Exo8KO mice were mated with the female GrphKO mice. Because the granuphilin and exophilin-8 genes are on the mouse X and 9 chromosomes, respectively, the resultant F1 generation is either male ($Grph^{-/Y}$, $Exo8^{+/-}$) or female ($Grph^{+/-}$, $Exo8^{+/-}$). By crossing these F1 mice, GE8DKO mice, as well as the WT, GrphKO, and Exo8KO mice, were generated in the F2 generation and used for experiments. Although the resultant F2 mice have a mixture of C57BL/6N and C3H/He genomes, we expected that significant phenotypic changes due to the loss of Rab27 effectors may be preserved despite any influences due to randomly distributed differences in the genome. In fact, we found similar differences in exocytic profiles between WT and Exo8KO cells both in the C57BL/6N background (*Figure 2*) and in the mixture of the C57BL/6N and C3H/He backgrounds (*Figure 6*). Furthermore, GrphKO cells in the mixture of the C57BL/6N and C3H/He backgrounds showed changes in granule localization and exocytosis (*Figure 6*), consistent with the reported phenotypes of GrphKO cells in the C3H/He background (*Gomi et al., 2005*). Pancreatic islet isolation and dissociation into monolayer cells and insulin secretion assays were performed as described previously (*Gomi et al., 2005*; *Wang et al., 2020*). Briefly, islets were isolated from cervically dislocated mice by pancreatic duct injection of collagenase solution, and size-matched five islets were cultured overnight in a 24-well plate. Monolayer islet cells were prepared by incubation with trypsin-EDTA solution and were cultured for further 2 days. Insulin released from isolated islets or monolayer cells was measured by an AlphaLISA insulin kit (Perkin-Elmer) or insulin high range and ultra-sensitive kits (Cisbio).

## DNA manipulation

Mouse cDNAs of granuphilin (*Wang et al., 1999*), exophilin-8 (*Mizuno et al., 2011*), and melanophilin (*Wang et al., 2020*) were cloned previously. Human cDNAs of SEC3, SEC5, SEC6, SEC8, SEC10, SEC15, EXO70, and EXO84 in the pEGFP-C3 vector were gifts from Dr. Channing J. Der (Addgene plasmid # 53755-53762; http://n2t.net/addgene:53755-53762; RRID:Addgene_53755-53762; *Martin et al., 2014*). Adenoviruses encoding insulin-EGFP and KuO-granuphilin were described previously (*Kasai et al., 2008*; *Mizuno et al., 2016*). Hemagglutinin (HA-), FLAG-, MEF-, One-STrEP-FLAG (OSF), and mCherry-tagged exophilin-8 and melanophilin were made previously (*Fan et al., 2017*; *Wang et al., 2020*). To express exogenous protein, HEK293A cells were transfected with the plasmids

using Lipofectamine 3000 reagent (Invitrogen), whereas MIN6 and primary pancreatic beta cells were infected with adenoviruses.

## Cell lines, antibodies, and immunoprocedures

MIN6 cells (RRID:CVCL_0431) were originally provided by Dr. Jun-ichi Miyazaki (Osaka University; *Miyazaki et al., 1990*). HEK293A cells (RRID:CVCL_6910) were purchased from Invitrogen (Cat# R70507). Guinea pig anti-insulin serum was a gift from H. Kobayashi (Gunma University). Rabbit polyclonal anti-exophilin-8 (αExo8N) and anti-granuphilin (αGrp-N) antibodies are described previously (*Fan et al., 2017*; *Yi et al., 2002*). mCherry nanobody was a gift from Drs. Y. Katoh and K. Nakayama (Kyoto University) (*Katoh et al., 2016*). The sources of commercially available antibodies and their concentrations used are listed in Key Resources Table. Cells were lysed by lysis buffer consisting of 50 mmol/L Tris-HCl, pH 7.5, 150 mmol/L NaCl, 10% (w/v) glycerol, 100 mmol/L NaF, 10 mmol/L ethylene glycol tetraacetic acid, 1 mmol/L $Na_3VO_4$, 1% Triton X-100, 5 µmol/L $ZnCl_2$, 1 mmol/L phenylmethylsulfonyl fluoride, and cOmplete Protease Inhibitor Cocktail (Roche). Immunoblotting and immunoprecipitation were performed as described previously (*Matsunaga et al., 2017*; *Wang et al., 2020*). VIP assay using the nanobody was performed as described previously (*Katoh et al., 2015*). Briefly, HEK293A cells on 10 cm dish were transfected with pEGFP-Sec and either pmCherry-Melanophilin or pmCherry-Exophilin-8 by Lipofectamine 3000. After 48 hr, cells were lysed by 1 mL of lysis buffer, and the cell lysate was centrifuged at 10,000 × *g* for 10 min, and the supernatant was subjected to immunoprecipitation with a 5 µL gel volume of mCherry nanobody-bound glutathione Sepharose. The beads were washed three times with lysis buffer and transferred to 35 mm glass base dishes (Glass φ12, IWAKI). Green and red fluorescence of beads was observed by confocal laser scanning microscopy. Acquisition of images was performed under fixed conditions. For immunofluorescence procedures, monolayer primary beta cells seeded at 3–5 × $10^4$ cells on a poly-L-lysine-coated glass base dish were cultured overnight in RPMI-1640 medium (11.1 mmol/L glucose) supplemented with 10% fetal calf serum and were fixed by 4% paraformaldehyde for 30 min at room temperature. The cells were washed by phosphate buffered saline (PBS) and permeabilized by PBS containing 0.1% Triton X-100 and 50 mmol/L $NH_4Cl$ for 30 min. After blocking with 1% bovine serum albumin in PBS for 30 min, the cells were immunostained and observed under confocal microscopy with a ×100 oil immersion objective lens (1.49 NA). Quantification of immunoblot signals was performed using ImageQuant TL software (Cytiva). Quantification of colocalization between two proteins was performed using NIS Element Viewer software (Nikon).

## TIRF microscopy

TIRF microscopy was performed as described previously (*Wang et al., 2020*). Briefly, monolayer islet cells on glass base dish were infected with adenovirus encoding insulin-EGFP. Two days thereafter, the cells were preincubated for 30 min in 2.8 mmol/L low glucose (LG)-containing Krebs-Ringer bicarbonate (KRB) buffer at 37°C. They were then incubated in 25 mmol/L high glucose (HG)-containing buffer for 20 min or 60 mmol/L high potassium buffer for 6 min. TIRF microscopy was performed on a ×100 oil immersion objective lens (1.49 NA). The penetration depth of the evanescent field was 100 nm. Images were acquired at 103 ms intervals. Fusion events with a flash followed by diffusion of EGFP signals were manually selected and assigned to one of three types: residents, which were visible for more than 10 s prior to fusion; visitors, which became visible within 10 s of fusion; and passengers, which were not visible prior to fusion. In case of coinfection with adenovirus encoding KuO-granuphilin, sequential multi-color TIRF microscopy was performed as described previously (*Mizuno et al., 2016*). Briefly, EGFP was excited using a 488 nm solid-state laser, whereas KuO was excited using a 561 nm laser. Excitation illumination was synchronously delivered from an acousto-optic tunable filter-controlled laser launch. A dual-band filter set (LF488/561A; Semrock) was applied on a light path.

## Mass spectrometry

MIN6 cells (2 × $10^7$ cells in ten 15 cm dishes) were infected with adenovirus encoding FLAG, FLAG-melanophilin, or MEF-exophilin-8. The anti-FLAG immunoprecipitates were subjected to gel electrophoresis and visualized by Oriole fluorescent gel staining (Bio-Rad). Specific bands were excised and digested in gels with trypsin, and the resulting peptide mixtures were analyzed by a LC-MS/MS

system, as described previously (*Matsunaga et al., 2017*). All MS/MS spectra were separated against the *Mus musculus* (mouse) proteome data set (UP000000589) at the Uniplot using Protein pilot software (SCIEX).

## Silencing of the exocyst component in mouse pancreatic islet cells

The On-Target plus Set of four siRNA against mouse Exoc5 (J-047583-11 and -12; Cat# 105504) and mouse Exoc4 (J-051541-12-0050; Cat# 20336), and the On-Target plus non-targeting pool siRNA were purchased from Horizon Discovery Ltd. Mouse pancreatic islet cells suspended in $1 \times 10^5$ cells/280 μL were transfected with 100 nmol/L siRNA using Lipofectamine RNAiMAX reagent (Invitrogen). After plating on a 24-well plate or glass base dish for 48 hr, the cells were transfected with the same siRNA for the second time. After another 48 hr, the cells were subjected to immunoblotting analyses, immunofluorescent staining, insulin secretion assays, or TIRF microscopy after infection with adenovirus encoding insulin-EGFP.

## Statistical analysis

All quantitative data were assessed as the mean ± SEM. p-Values were calculated using Student's *t* test or a one-way ANOVA with a Tukey multiple-comparison using GraphPad Prism software.

## Data and resource availability

All data generated or analyzed during this study are included in the manuscript and supporting files. All noncommercially available resources generated and/or analyzed during this study are available from the corresponding author upon reasonable request.

## Acknowledgements

The authors thank T Nara, E Kobayashi, and T Ushigome for colony maintenance of mice, and S Shigoka for preparing the manuscript. This work was supported by Japan Society for the Promotion of Science KAKENHI grants JP19H03449 to TI, JP20K06535 to KMi, and JP20K15742 to HW. It was also supported by grants from Eli Lilly Japan Donation and Teijin Pharma Donation to TI. This work was the result of using research equipment shared in the MEXT Project for promoting public utilization of advanced research infrastructure (program for supporting the introduction of the new sharing system) grant number JPMXS0420600122.

## Additional information

### Funding

| Funder | Grant reference number | Author |
| --- | --- | --- |
| Japan Society for the Promotion of Science | JP19H03449 | Tetsuro Izumi |
| Japan Society for the Promotion of Science | JP20K06535 | Kouichi Mizuno |
| Japan Society for the Promotion of Science | JP20K15742 | Hao Wang |
| Eli Lilly Japan | | Tetsuro Izumi |
| Teijin Pharma | | Tetsuro Izumi |

The funders had no role in study design, data collection and interpretation, or the decision to submit the work for publication.

### Author contributions

Kunli Zhao, Data curation, Formal analysis, Validation, Investigation; Kohichi Matsunaga, Data curation, Formal analysis, Supervision, Validation, Investigation, Methodology; Kouichi Mizuno, Supervision, Funding acquisition, Validation, Methodology; Hao Wang, Supervision, Funding acquisition,

Methodology; Katsuhide Okunishi, Resources; Tetsuro Izumi, Conceptualization, Data curation, Supervision, Funding acquisition, Validation, Visualization, Writing - original draft, Project administration, Writing - review and editing

## Author ORCIDs
Tetsuro Izumi ⓘ http://orcid.org/0000-0002-0974-7384

## Ethics
Animal experiments were performed according to the rules and regulations of the Animal Care and Experimental Committees of Gunma University (permit number: 22-010; Maebashi, Japan).

## Decision letter and Author response
Decision letter https://doi.org/10.7554/eLife.82821.sa1
Author response https://doi.org/10.7554/eLife.82821.sa2

---

## Additional files

### Supplementary files
• MDAR checklist

### Data availability
All data generated or analyzed during this study are included in the manuscript and supporting files.

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
