## [Editor Report]

This study examines how Rab27 and its different effectors regulate insulin secretory granule exocytosis. Using single and double knockouts and rescue experiments, the work presents convincing data to characterize the relative hierarchy of the Rab27 effectors, their action with the exocyst, and their roles in distinct types of exocytosis. Overall, this is a valuable study that sheds new light on the regulation of distinct forms of secretory granule exocytosis.

---

## [Decision Letter]

**Decision letter after peer review:**

Thank you for submitting your article "Functional hierarchy among different Rab27 effectors involved in secretory granule exocytosis" for consideration by *eLife*. Your article has been reviewed by 3 peer reviewers, and the evaluation has been overseen by a Reviewing Editor and Anna Akhmanova as the Senior Editor. The following individuals involved in review of your submission have agreed to reveal their identity: Jing Hughes (Reviewer #2); Alistair Hume (Reviewer #3).

Essential revisions:

1) Metabolic phenotyping of the DKO mouse needs to be performed, ideally including dynamic measurements of insulin secretion (Reviewer 2)

2) Additional data are require to strengthen the connection between the exocyst and the Rab27 effectors by additional loss of function experiments combined with live cell imaging of the components. It would also be important to clarify how melanophilin expression is affected by Exo8 KO (Reviewer 1).

3) The manuscript should use more precise language, noting the limitations of the interpretation and providing alternate explanations, particularly for the possibility that another pathway(s) for insulin secretion exists in these preparations (Reviewer 3).

*Reviewer #1 (Recommendations for the authors):*

1) The authors show both biochemically and by granule imaging that GSIS is reduced from β cells of exophilin-8 and melanophilin KO mice. However, information about granule density at the plasma membrane and insulin content in the cells is lacking. It is therefore difficult to know whether the secretory defect can also be explained by a relative insulin deficiency.

2) A problem with the study is that the authors follow the dynamic process of insulin secretion by live cell TIRF microscopy of glucose-stimulated cells, but then look at the localization of the Rab27 effectors by immunofluorescence staining of fixed cells, where the culture conditions at the time-point of fixation are unknown (e.g. glucose concentration). This makes it difficult to directly compare the results obtained by these two methods. The paper would be much strengthened from live cell imaging of Exophilin-8, melanophilin, granuphilin (as is done elegantly in Figure 6) and components of the exocyst complex. Such experiments will allow the authors to draw much more accurate conclusions about the identity of different granule subpopulation and how they relate to the different Rab27 effectors. This becomes particularly important for the exocyst components, where the authors data indicate that these would be present on a majority of granules that undergo exocytosis.

3) The authors show that melanophilin levels are reduced by 50% in exophilin-8 KO β cells, and suggest based on this that Exo8 may be required for stabilization of Melanophilin. This however, complicates the interpretation of the data in this model. How can the authors know that part of the exophilin-8 KO phenotype is not due to the simultaneous reduction in melanophilin expression? Although the authors show that the KO phenotype can be rescued by re-expression of exo-8, they don't show what happens with melanophilin expression under these conditions. It would be essential to show this, and also to test to what extent melanophilin re-expression affect the phenotype of the Exo-8 KO cells.

4) In figure 1 – supplement 1 the authors show a strikingly high degree of colocalization between exo-8 and Mlph. However, based on functional data and on the model shown in Figure 7, these two proteins would only be expected to colocalize on a small subset of granules that immediately fuse with the plasma membrane. What is the reason for this discrepancy?

5) The authors demonstrate, through biochemical assays, interactions between the exocyst complex and both Exo-8 and Mlph. Based on results from IP-experiments, the authors conclude that Exo-8 interacts with subcomplex I via Sec8 and that Mlph interacts with subcomplex 2 via Exo70. These are interesting results that corroborates earlier studies indicating a link between insulin granule mobilization/secretion and the exocyst complex.

However, it is very difficult to envisage how this interaction plays into the regulation of granule mobilization and exocytosis. In particular, it is unclear how the interaction with the exocyst complex affects the localization of the Rab27 effectors and what its role is in Grph-dependent exocytosis. The evidence for the Rab27 effectors associating with the exocyst complex would be strengthened by showing the loss of effectors from Rab27 positive granules following knockdown of Sec8 (for Exo-8) and Exo70 (for Mlph) and showing that the interaction between Exo-8 and Mlph is lost following KD by IP. It would also be relevant to determine to what extent Grph-dependent exocytosis also require the Exocyst complex (in the model it does not, but this is not experimentally tested). To show colocalization with Sec6 as is done now (Figure 3A,B) is not appropriate since it is unclear what granule pool is labelled by this protein and whether its localization is affected by Sec10 KD. In fact, it is unclear to me why the authors chose to reduce the expression of Sec10 and not Sec8/Exo70, which might have allowed the authors to more specifically dissect the Exo-8/Mlph pathways.

6) The components of the exocyst complex can have functions independent of the exocyst. Since the authors only reduce the expression of Sec10, it may be difficult to draw conclusions about the entire complex. More extensive loss of function studies would be required or at a minimum this possibility should be discussed.

7) The authors state that exophilin-8 and melanophilin were re-expressed at the endogenous level (Figure 1B), which is also what the Western blot shows. However, re-expression of exophilin-8 results in a much larger proportion of cells with peripheral granule accumulation (compare Figure 1A to Figure 1B; 45% and 65%). A likely explanation for this is that not all cells are transduced with the adenovirus. It would therefore be better to not say at the endogenous level but instead just talk about re-expression.

8) Could the authors explain the rational for the choice of glucose concentrations when stimulating the β cells. 2.8 mM is sub-physiological and 25 mM is supra-physiological. More reasonable would have been to increase the glucose concentration from 5 mM to 10 mM.

9) It is shown that exo-8 is not involved in Grph-mediated exocytosis of docked granules but still somehow involved in exocytosis of docked granules. How can Exo-8 deficiency primarily influence the release of granules that are already docked at the plasma membrane? Based on the proposed mechanism, Exo-8 would be expected to primarily influence exocytosis of granules that are mobilized from the cytosol along actin filaments.

10) It would be appropriate to also include Grph KO and WT cells in the experiments presented in Figure 6, so that one can compare the degree of rescue obtained.

11) How do the authors propose that actin is involved? Is there an actin-independent (granuphilin) and actin-dependent (melanophilin) form of exocytosis in β cells?

12) Quantify the degree of colocalization between Sec6 and insulin in Figure 3 —figure supplement 1.

13) Cropped images in figure 1A and 1B are not correctly cropped.

14) The small decrease in Pearson correlation coefficient reported in Figure 3C should not be artificially inflated by y-axis cropping.

15) The authors several times talk about "immediate" binding. Could they please clarify what they mean by this. Is it direct binding?

16) The authors state that GSIS from the GE8DKO islets exhibit an intermediate level between GrphKO and Exo8KO, but this is not what is shown in Figure 5A (there are no differences at all).

17) In Figure 5C, Exo-8 KO is without effect on insulin secretion visualized by granule imaging (total and resident granules). This is different from other experiments and complicates interpretations also from the GE8DKO cells. Can the authors please comment on the reason for this.

18) How can the authors know that they are recording exocytic events from β cells and not from other islet cell types?

*Reviewer #2 (Recommendations for the authors):*

Suggestions for additional experiments or clarifications:

– While Exo8 and melanophilin loss-of-function mouse models have been previously characterized (Fan et al. 2017, Wang et al. 2020), the metabolic phenotype of double knockout ME8DKO mice is not presented here. It would be informative to include some standard in vivo metabolic testing such as IPGTT, body morphometrics, fasting glucose and diabetes incidence on chow or challenge diets, to see if these mice phenocopy Exo8KO and what does the loss of this pathway mean for diabetes propensity.

– Dynamic insulin secretion as the authors have previously performed (again, in Fan et al. 2017 and Wang et al. 2020) would complement the static secretion data in Figure 1C by showing whether the secretory defect lies in 1st or 2nd phase, and how this might correspond to the release of different insulin pools – resident, passenger, or visitor.

– The observation that melanophilin-only knockouts (MlphKO) have a milder phenotype than Exo8KO or ME8DKO suggests that there may be effectors other than melanophilin downstream of Exo8. Now that we have double KO data, what are the authors' conclusion regarding role of melanophilin, is it redundant and therefore expendable for Exo8 or Rab27 function? Under what physiologic conditions might this melanophilin-mediated, passenger-type exocytosis pathway be important?

*Reviewer #3 (Recommendations for the authors):*

In this manuscript Zhao et al. investigated how multiple Rab27 effectors work to regulate insulin secretion by murine pancreatic b-cells. They do this by comparing the phenotypes of b-cells/islets lacking effectors doubly or singly. Their main findings/contributions are that

1. Mlph works downstream of Myrip/exophilin-8 to mobilise granules for fusion from the actin network to the plasma membrane after stimulation.

2. Mlph and exophilin-8 interact via the exocyst

3. Down-regulation of exocyst affects exocytosis in cells expressing exophilin-8

4. Exophilin-8 promotes fusion of granules docked by granuphilin at the membrane

5. Exophilin-8 not required for Grph related granule docking at the plasma membrane

A model for how the three effectors coordinate ISG secretion. According to this model there are 2 insulin secretion pathways in b-cells; (a) where Exo8 acts upstream of Mlph and with actin/Myosin Va/VIIa, exocyst and syntaxin 4 to move dock granules in actin and promote exocytosis, and (b) where Exo8 works in an antagonistic manner with Grph promoting secretion of granules docked at the membrane by Grph.

In support of 1 they show that:

insulin granules are docked at the cortex (a prelude to fusion) in WT and MlphKO, but not Exo8KO and double ME8DKO b-cells (Figure 1).

In ME8DKO cells they show that expression of Exo8 but not Mlph restores cortical distribution of granules (Figure 1).

Using TIRFM analysis of insulin secretion in pancreatic b-cells that fusion of non-resident granules (passenger and visitor types) is reduced in MlphKO cells compared with controls in response to stimulation (Figure 1).

In support of 2 they show that:

over-expressed/tagged and endogenous Exo8 and mlph can be co-precipitated in MIN6 cells (Figure 2).

exocyst components Sec6 and Sec10 co-IP with Mlph and Exo8 (Figure 2).

Depletion of sec10 reduces Mlph association with sec6 positive insulin granules (Figure 3).

However, it would be important to show this directly e.g., by showing the results of Mlph/Exo8 IP in exocyst depleted cells to confirm that Mlph/Exo8 interaction is via the exocyst.

In support of 3 they show that:

using ELISA assays that glucose dependent insulin secretion is reduced in wild-type, but not Exo8KO, cells when Sec10 is KD (Figure 4A).

In support of 4 they show that:

insulin secretion in double granuphilin (Grph)/Exo8 DKO b-cells is reduced compared with single Grph KO cells (Figure 5A). Related to this they show that granule number in the TIRF field is affected GrphKO and GE8DKO but not Exo8KO. This seems surprising given that in Figure 1 the authors show that the number of docked granules is increased in Exo8KO cells.

In support of 5

they use TIRFM to show that granules are docked at the membrane by Exo8KO but not GrphKO or GE8DKO (Figure 5B).

Overall, their data support the model shown in Figure 6.

This is an interesting study but leaves open some important questions regarding insulin secretion and the role of Rab27a and its effectors in this process.

The finding that GE8DKO islets release more insulin than wild-type (Figure 5A) is striking and suggests that another pathway(s) for insulin secretion exists in these preparations. Can the authors discuss the possible molecular mechanism of this pathway in their manuscript/include this in their model to present a more complete picture of the mechanisms of granule exocytosis?

Related to this while I understand that it may be out of the remit of this project to characterise Grph/Exo8 independent pathways from scratch it would nevertheless be highly relevant to this study on Rab27a effectors for the authors to investigate the possibility that Rab27 and Mlph are involved in this pathway. For example, by generating triple Mlph, Exo8 and Grph8 KO islets or depleting in Mlph in GE8DKO islets and examining the effect of this on exocytosis and how this relates to exocytosis in the absence of Rab27a.

Regarding the possible different pathways have the authors noticed any differences in the kinetic profile of fusion events as monitored by TIRFM this might provide interesting insight into the relationship and mechanism between the proposed different pathways.

---

## [Author Response]

Essential revisions:1) Metabolic phenotyping of the DKO mouse needs to be performed, ideally including dynamic measurements of insulin secretion (Reviewer 2)

We have performed the above experiments and presented the data in Figure 2B, 2E, and Figure 2—figure supplement 1.

2) Additional data are require to strengthen the connection between the exocyst and the Rab27 effectors by additional loss of function experiments combined with live cell imaging of the components. It would also be important to clarify how melanophilin expression is affected by Exo8 KO (Reviewer 1).

We have performed additional loss of function experiments combined with live cell imaging of the components and presented the data in Figure 7 and Figure 7—figure supplement 1. We found differential roles in the resident type exocytosis between exophilin-8 and the exocyst. Please see our responses to Reviewer 1’s comments 2) and 5). Furthermore, we clarify the effect of melanophilin expression in Exo8KO cells by performing experiments in heterozygous MlphKO cells (Figure 3—figure supplement 1). Please see our response to Reviewer 1’s comment 3).

3) The manuscript should use more precise language, noting the limitations of the interpretation and providing alternate explanations, particularly for the possibility that another pathway(s) for insulin secretion exists in these preparations (Reviewer 3).

We have asked a professional external editor in the US to correct English and reflected the changes in a revised manuscript. Furthermore, we have added description about the existence of another pathway in the last part of Discussion.

Reviewer #1 (Recommendations for the authors):1) The authors show both biochemically and by granule imaging that GSIS is reduced from β cells of exophilin-8 and melanophilin KO mice. However, information about granule density at the plasma membrane and insulin content in the cells is lacking. It is therefore difficult to know whether the secretory defect can also be explained by a relative insulin deficiency.

We have added the data of insulin content in Figure 2—figure supplement 1B and presented the data of granule density in Figure 2C.

2) A problem with the study is that the authors follow the dynamic process of insulin secretion by live cell TIRF microscopy of glucose-stimulated cells, but then look at the localization of the Rab27 effectors by immunofluorescence staining of fixed cells, where the culture conditions at the time-point of fixation are unknown (e.g. glucose concentration). This makes it difficult to directly compare the results obtained by these two methods. The paper would be much strengthened from live cell imaging of Exophilin-8, melanophilin, granuphilin (as is done elegantly in Figure 6) and components of the exocyst complex. Such experiments will allow the authors to draw much more accurate conclusions about the identity of different granule subpopulation and how they relate to the different Rab27 effectors. This becomes particularly important for the exocyst components, where the authors data indicate that these would be present on a majority of granules that undergo exocytosis.

We have described in Materials and methods that monolayer primary β cells were cultured in RPMI-1640 medium (11.1 mmol/L glucose) before the fixation.

About the identity of different granule subpopulation related to the different Rab27 effectors, we would first like the reviewer to understand that, although each Rab27 effector regulates a distinct exocytic step, all Rab27 effectors previously examined in pancreatic β cells, including Noc2 that also interacts with Rab2a located in the perinuclear region (Matsunaga, J Cell Sci, 2017), are not differentially located on specific granules where they function, but are distributed on almost all insulin granules throughout the cytoplasm via interaction with multiple Rab27a molecules on single granules (see granuphilin for Yi et al., Mol Cell Biol; exophilin-8 for Fan et al., *ELife*, 2017; and melanophilin for Wang et al., Diabetes, 2020). For example, we previously showed by both TIRF and confocal microscopy that granuphilin and melanophilin, which should separately mediate the resident type and the passenger type, respectively, are colocalized on granules (Supplementary Figure 6 in Wang et al., Diabetes, 2020). Thus, each effector is not necessarily located on different granule subpopulation. As discussed previously (Izumi, Cell Struct Funct, 2021), the fate of each individual granule is not defined by the presence of the particular Rab27 effector per se but appears to be stochastically determined by its interactions with specific proteins and membrane lipids in a local milieu around the granule. As the reviewer noticed, the exocyst component, Sec6, is also distributed to almost all the granules (Figure 4).

For live cell imaging, it is very important to express exogenous fluorescent protein in a physiological condition. For example, overexpressed granuphilin changes granule distribution (Torii et al., J Biol Chem, 2004) and severely inhibits granule exocytosis (Torii et al., Mol Cell Biol, 2002). Furthermore, we could not perform live cell imaging previously in the case of melanophilin, because mCherry-melanophilin expressed in the presence of endogenous melanophilin was not located on granules, but aberrantly located along the actin microfilaments, and because its expression at the endogenous level in melanophilin-deficient cells display too weak fluorescence to monitor granule exocytosis (please see the details in Wang et al., Diabetes, 2020). Therefore, we always had to express fluorescent Rab27 effectors in the absence of endogenous proteins. This is thought to be particularly critical for the exocyst that generally functions as a heterooctameric protein complex, because exogenous expression of one fluorescent component unbalances the quantities among the components. In support of this view, we found marked effects of SEC10 knockdown on granule exocytosis despite the partial downregulation (Figure 4—figure supplement 1A). Therefore, decent works involving live cell imaging of the exocyst have been performed in knockin cells in both yeast and mammalian cells (see examples, Boyd et al., J Cell Biol 2004, Donovan et al. J Cell Biol, 2015; Ahmed et al., Nat Commun, 2018). Given that our data in Figures 3 and 4 suggest that the holo-exocyst connects only a part of exophilin-8 and melanophilin, we do not want to perform experiments overexpressing one component of the exocyst, which unlikely functions as a good marker of the holo-exocyst and likely yields uncertain and possibly artificial results. We would like to do live cell imaging of the exocyst in primary β cells, after we generate the knockin mouse for one of the components in the future study. We hope that the editors and the reviewers understand that the main purpose of this study is to identify the functional relationship among different Rab27 effectors.

We also would like to point out that exophilin-8 and the exocyst are anyway involved in both resident and passenger types of exocytosis, as shown in Figures 2D and 5C, and therefore, their live-cell imagimg would not identify different granule subpopulation mediating the specific type of exocytosis. It should also be noted that TIRF microscopy does not provide direct information in the cell interior. For example, we could not get information where granules mediating the passenger exocytosis, which appear only in one frame prior to fusion, have been originally located before stimulation.

However, in the case of granuphilin that molecularly tether granules to the plasma membrane, live cell imaging by TIRF microscopy generates useful information. Given that the evanescent field illuminates fluorescence within 100-200 nm of the coverslip and that the radius of insulin granules is 150-200 nm, granuphilin-positive granules should harbor granuphilin in their hemisphere that is closer to the plasma membrane, and so are likely physically tethered to the plasma membrane via interaction with the closed form of syntaxins. In fact, we previously found that granules visible under TIRF microscopy can be divided into two classes: granuphilin-positive, immobile granules and granuphilin-negative, mobile granules (Mizuno et al., Sci Rep, 2016). Therefore, we can discriminate granules physically tethered to the plasma membrane and those just locating nearby. Please also understand that granuphilin-negative granules under TIRF microscopy mean its absence in their hemisphere that is closer to the plasma membrane but may still harbor granuphilin in their opposite hemisphere. To investigate the role of the exocyst in the resident type exocytosis, we performed new experiments in response to the reviewer’s point 5 below.

3) The authors show that melanophilin levels are reduced by 50% in exophilin-8 KO β cells, and suggest based on this that Exo8 may be required for stabilization of Melanophilin. This however, complicates the interpretation of the data in this model. How can the authors know that part of the exophilin-8 KO phenotype is not due to the simultaneous reduction in melanophilin expression? Although the authors show that the KO phenotype can be rescued by re-expression of exo-8, they don't show what happens with melanophilin expression under these conditions. It would be essential to show this, and also to test to what extent melanophilin re-expression affect the phenotype of the Exo-8 KO cells.

Although melanophilin must be kept at the endogenous level as described above, it is not easy to exactly compensate the decreased amount of endogenous melanophilin in each Exo8KO cell. Therefore, we instead examined the heterozygous MlphKO cells that may express it at a half level of the endogenous melanophilin. Indeed, the level of endogenous melanophilin in the heterozygous cells was 65.4 ± 4.7% of that in WT cells, which was equivalent to the level in Exo8KO cells (59.8 ± 9.0%). The finding that these heterozygous cells did not show reduced insulin secretion (Figure 3—figure supplement 1) suggests that the decrease in melanophilin expression is unlikely responsible for the reduced insulin secretion in Exo8KO cells.

4) In figure 1 – supplement 1 the authors show a strikingly high degree of colocalization between exo-8 and Mlph. However, based on functional data and on the model shown in Figure 7, these two proteins would only be expected to colocalize on a small subset of granules that immediately fuse with the plasma membrane. What is the reason for this discrepancy?

As described above in response to the reviewer’s point 2, the localization of each Rab27 effector is not restricted to specific granules. The localization of Rab27 effectors is determined via interaction with multiple Rab27 molecules existing on granules. For example, we previously showed the melanophilin E14A mutant deficient in Rab27 binding are not colocalized with granules at all (Wang et al., Diabetes, 2020). Therefore, the two proteins are highly colocalized on each insulin granule, although a subset of these proteins are expected to interact with, as the reviewer points out.

5) The authors demonstrate, through biochemical assays, interactions between the exocyst complex and both Exo-8 and Mlph. Based on results from IP-experiments, the authors conclude that Exo-8 interacts with subcomplex I via Sec8 and that Mlph interacts with subcomplex 2 via Exo70. These are interesting results that corroborates earlier studies indicating a link between insulin granule mobilization/secretion and the exocyst complex.However, it is very difficult to envisage how this interaction plays into the regulation of granule mobilization and exocytosis. In particular, it is unclear how the interaction with the exocyst complex affects the localization of the Rab27 effectors and what its role is in Grph-dependent exocytosis. The evidence for the Rab27 effectors associating with the exocyst complex would be strengthened by showing the loss of effectors from Rab27 positive granules following knockdown of Sec8 (for Exo-8) and Exo70 (for Mlph) and showing that the interaction between Exo-8 and Mlph is lost following KD by IP. It would also be relevant to determine to what extent Grph-dependent exocytosis also require the Exocyst complex (in the model it does not, but this is not experimentally tested). To show colocalization with Sec6 as is done now (Figure 3A,B) is not appropriate since it is unclear what granule pool is labelled by this protein and whether its localization is affected by Sec10 KD. In fact, it is unclear to me why the authors chose to reduce the expression of Sec10 and not Sec8/Exo70, which might have allowed the authors to more specifically dissect the Exo-8/Mlph pathways.

more specifically dissect the Exo-8/Mlph pathways.

In response to the reviewer’s question, we first examined the effects of SEC10 knockdown on the intracellular localization of exophilin-8 and melanophilin, and found that SEC10 knockdown does not affect their colocalization with insulin or Rab27a (Figure 4—figure supplement 1). It is not surprising, because Rab27 effectors is localized on granules via interaction with Rab27, as described above. Therefore, it is not expected that any effectors will be lost from Rab27 positive granules even if other exocyst component is downregulated.

The reason we select SEC10 to downregulate the exocyst is that this component plays a key role to form the holo-exocyst, because previous VIP assays indicate that the two subcomplexes are biochemically linked via the interactions between Sec10 in the subcomplex 2 and Sec8 and/or Sec3 in the subcomplex 1 (Katoh et al., J Cell Sci, 2015). If SEC10 knockdown disrupts the interaction between exophilin-8 and melanophilin, the holo-exocyst likely links the two effectors. Consistent with this view, SEC10 knockdown partially disrupted the colocalization with SEC6 of melanophilin, but not that of exophilin-8 (Figure 4A).

However, according to the reviewer’s suggestion, we also tried to knockdown SEC8 and EXO70. The available siRNAs downregulated SEC8 significantly (Figure 4—figure supplement 1A), but unfortunately not EXO70 (see Author response image 1). In fact, siRNAs against EXO70 was reported to be inefficient (Ahmed et al., Nat Commun, 2018). We thus investigated whether the interaction between exophilin-8 and melanophilin is lost following SEC8 KD by IP, and found that it markedly disrupted the complex formation (Figure 4B), which further supports the involvement of the exocyst in the interaction of the two effectors.

**Author response image 1. sa2fig1:** 

In response to the reviewer’s question “what the role of the exocyst is in Grph-dependent exocytosis”, we first performed TIRF experiments in GrphKO cells. SEC10 knockdown did not further decrease the number of visible granules in GrphKO cells (Figure 7A), as found in GE8DKO cells (Figure 6B). However, it markedly decreased the resident type exocytosis (Figure 7B), in contrast to the case in GE8DKO cells (Figure 6C). To directly assess the influence of exocyst deficiency on granuphilin-mediated, docked granule exocytosis, we then expressed Kuo-Grph in GrphKO cells, as performed in Figure 6D,E. As shown in Figure 7C,D, SEC10 knockdown specifically decreased the fusion probability of granuphilin-negative granules (Figure 7D), in contrast that exophilin-8 deficiency markedly decreased that of granuphilin-positive granules (Figure 6E). As described in Discussion, these data suggest that the exocyst promote the exocytosis of granuphilin-negative, mobile granules by tethering them to the plasma membrane, as found in other constitutive secretory pathways. In contrast, exophilin-8 appears to promote fusion of granuphilin-mediated docked (tethered) granules by converting the associated, closed form of syntaxins into the open form.

Although we could not measure glucose-induced insulin secretion in the above experiments, we hope that the editors and the reviewers understand the vulnerable nature of primary islet cells after isolation. To our knowledge, there is no reports to do such multiple interventions, two times transfections and two kinds of adenovirus infections, for live cell imaging of primary β cells. In general, it is difficult to monitor two kinds of fluorescent proteins in addition to insulin-EGFP to examine their function in granule exocytosis in living β cells, without generating the knockin mouse that expresses multiple fluorescent protein instead of endogenous proteins.

6) The components of the exocyst complex can have functions independent of the exocyst. Since the authors only reduce the expression of Sec10, it may be difficult to draw conclusions about the entire complex. More extensive loss of function studies would be required or at a minimum this possibility should be discussed.

As described in response to the reviewer’s comment 5), we chose SEC10 knockdown to examine the involvement of the holo-exocyst, and performed additional experiments using SEC8 siRNA (Figure 4B). To our knowledge, although the exocyst has been investigated more than quarter century, the literature, including live cell imaging of the exocyst (Ahmad et al., Nat Commun, 2018), indicates that it functions as the holo-complex in cells.

7) The authors state that exophilin-8 and melanophilin were re-expressed at the endogenous level (Figure 1B), which is also what the Western blot shows. However, re-expression of exophilin-8 results in a much larger proportion of cells with peripheral granule accumulation (compare Figure 1A to Figure 1B; 45% and 65%). A likely explanation for this is that not all cells are transduced with the adenovirus. It would therefore be better to not say at the endogenous level but instead just talk about re-expression.

It is inevitable that adenovirus infection is somewhat inhomogeneous. We just meant the endogenous level in average.

8) Could the authors explain the rational for the choice of glucose concentrations when stimulating the β cells. 2.8 mM is sub-physiological and 25 mM is supra-physiological. More reasonable would have been to increase the glucose concentration from 5 mM to 10 mM.

We simply thought that the stronger stimulation makes the phenotypic differences more evident in the mutant cells. The glucose stimulation from 5 or 10 mM may be weak to detect significant differences.

9) It is shown that exo-8 is not involved in Grph-mediated exocytosis of docked granules but still somehow involved in exocytosis of docked granules. How can Exo-8 deficiency primarily influence the release of granules that are already docked at the plasma membrane? Based on the proposed mechanism, Exo-8 would be expected to primarily influence exocytosis of granules that are mobilized from the cytosol along actin filaments.

No, we did state that exophilin-8 is involved in Grph-mediated exocytosis, as shown in Figure 6. We also proposed that Exo8 promotes the priming reaction to remove a Grph-mediated exocytic clamp by associated proteins such as RIM-BP2, RIM, and Munc13. To support this view, we show that the absence of exophilin-8 inhibits the fusion of granuphilin-positive, docked granules more strongly compared with granuphilin-negative, undocked granules (Figure 6D,E). At present, we have no evidence that Exo8 primarily influences exocytosis of granules that are mobilized from the cytosol along actin filaments, because, under TIRF microscopy, the number of visible granules before stimulation is not changed by exophilin-8 deficiency (Figures 2C, 6B).

10) It would be appropriate to also include Grph KO and WT cells in the experiments presented in Figure 6, so that one can compare the degree of rescue obtained.

These experiments are performed to discriminate the effect of exophilin-8 deficiency on granuphilin-positive, docked granules from that on granuphilin-negative, untethered granules. We adjusted the expression level of granuphilin at endogenous level, as shown in Figure 6—figure supplement 1. Furthermore, the actual numbers of fusion events in WT and Exo8KO cells (Figure 6B) are equivalent to those found in mimic WT and mimic Exo8KO cells (Figure 6D).

11) How do the authors propose that actin is involved? Is there an actin-independent (granuphilin) and actin-dependent (melanophilin) form of exocytosis in β cells?

We should discriminate the role of actin tracks from that of the actin cortex in vesicle transport. For example, granules may reach the plasma membrane by generating actin tracks by another Rab27 effector, SPIRE, as shown in skin melanocytes (Alzahofi et al., Nat Commun, 2020). Based on the data in Figure 2, we suggest that melanophilin-mediated passenger type exocytosis is derived from granules captured within the actin cortex by Exophilin-8. Consistently, the increase in the passenger type exocytosis in the absence of granuphilin is markedly reduced to the level found in Exo8KO cells (Figure 6C). However, as shown in Figures 2C and 6B, the absence of exophilin-8 does not affect the number of granules visible under TIRF microscopy, which suggests that granules residing close to the membrane are not necessarily derived from granules anchored in the actin cortex.

12) Quantify the degree of colocalization between Sec6 and insulin in Figure 3 —figure supplement 1.

We have removed this data, because the reviewer mentioned in the comment 5) that Sec6 is not appropriate since it is unclear what granule pool is labelled by this protein and whether its localization is affected by Sec10 KD. We instead examined the effects of SEC10 knockdown on the granule localization of exophilin-8 and melanophilin, and quantified the degree of colocalization (Figure 4—figure supplement 1).

13) Cropped images in figure 1A and 1B are not correctly cropped.

Although the images were correctly cropped in the original file, they appear to be improperly cropped during the conversion process to the PDF file. We have corrected them.

14) The small decrease in Pearson correlation coefficient reported in Figure 3C should not be artificially inflated by y-axis cropping.

We have corrected the y-axis in Figure 4A (original Figure 3C).

15) The authors several times talk about "immediate" binding. Could they please clarify what they mean by this. Is it direct binding?

In general, coimmunoprecipitation experiments do not prove direct binding. However, visible IP assays show that only one of the exocyst components can interacts with exophilin-8 or melanophilin, and therefore we used the word “immediate”.

16) The authors state that GSIS from the GE8DKO islets exhibit an intermediate level between GrphKO and Exo8KO, but this is not what is shown in Figure 5A (there are no differences at all).

We corrected the sentence as below:

GE8DKO cells also exhibited an increase in GSIS compared with Exo8KO cells.

17) In Figure 5C, Exo-8 KO is without effect on insulin secretion visualized by granule imaging (total and resident granules). This is different from other experiments and complicates interpretations also from the GE8DKO cells. Can the authors please comment on the reason for this.

Compared with the experiments of Figure 2D that were performed in mice with the same B6 genetic background, the experiments in Figure 6C (original Figure 5C) were performed in mice with a mixture of B6 and C3H genetic backgrounds (see Materials and methods), which may have increased variance of the data and decreased the significance in differences. However, the mean values and differences between WT and Exo8KO cells are similar between these studies.

18) How can the authors know that they are recording exocytic events from β cells and not from other islet cell types?

Although we and others performing similar TIRF microscopic experiments did not discriminate β cells from other cells beforehand, we previously showed that most (90%) of the monolayer cells from pancreatic islets are β cells. See the examples of WT and melanophilin-deficient cells (Wang et al., Diabetes 2020). Furthermore, we did not include the data of cells that did not respond at all to glucose stimulation, which may be non-β cells or become inert during preparation and/or incubation.

Reviewer #2 (Recommendations for the authors):Suggestions for additional experiments or clarifications:– While Exo8 and melanophilin loss-of-function mouse models have been previously characterized (Fan et al. 2017, Wang et al. 2020), the metabolic phenotype of double knockout ME8DKO mice is not presented here. It would be informative to include some standard in vivo metabolic testing such as IPGTT, body morphometrics, fasting glucose and diabetes incidence on chow or challenge diets, to see if these mice phenocopy Exo8KO and what does the loss of this pathway mean for diabetes propensity.

We compared the metabolic phenotypes, such as body weight, blood glucose concentrations during IPGTT and those during ITT including fasting glucose concentrations among WT, MlphKO, Exo8KO, and ME8DKO mice. Each mouse strain displayed correlated insulin secretory capacities in batch (Figure 2A), islet perifusion (Figure 2B; Figure 2—figure supplement 1C), and TIRF microscopic assays (Figure 2D,E; Figure 2—figure supplement 1D). However, it should be noted that, because these effectors are widely and differentially expressed in other endocrine cells, their deficiency may differently affect secretion of other hormones influencing glucose tolerance, such as incretin and glucagon.

– Dynamic insulin secretion as the authors have previously performed (again, in Fan et al. 2017 and Wang et al. 2020) would complement the static secretion data in Figure 1C by showing whether the secretory defect lies in 1st or 2nd phase, and how this might correspond to the release of different insulin pools – resident, passenger, or visitor.

We have performed islet perifusion experiments (Figure 2—figure supplement 1C) and compared the time course of insulin secretion with that of granule exocytosis under TIRF microscopy (Figure 2—figure supplement 1D). The passenger type exocytosis is particularly decreased in a later 2nd phase in mutant cells (Figure 2E). However, the resident type exocytosis also tended to decrease in a later phase (Figure 2E).

– The observation that melanophilin-only knockouts (MlphKO) have a milder phenotype than Exo8KO or ME8DKO suggests that there may be effectors other than melanophilin downstream of Exo8. Now that we have double KO data, what are the authors' conclusion regarding role of melanophilin, is it redundant and therefore expendable for Exo8 or Rab27 function? Under what physiologic conditions might this melanophilin-mediated, passenger-type exocytosis pathway be important?

As shown in Figures 2A and 2D, MlphKO cells have a milder phenotype compared with Exo8KO or ME8DKO cells, because of the lack of decrease in the resident type exocytosis. However, melanophilin deficiency induces similar decreases in the passenger type exocytosis whether exophilin-8 is present (MlphKO cells) or not (ME8DKO cells), which indicates that melanophilin does not merely play a redundant role, but a unique role in this type of exocytosis. In support of this view, both effectors function through different protein complexes, RIM-BP2-myosin-VIIa and myosin-Va-syntaxin-4, respectively (Fan et al., 2017 and Wang et al., 2020).

Reviewer #3 (Recommendations for the authors):In this manuscript Zhao et al. investigated how multiple Rab27 effectors work to regulate insulin secretion by murine pancreatic b-cells. They do this by comparing the phenotypes of b-cells/islets lacking effectors doubly or singly. Their main findings/contributions are that1. Mlph works downstream of Myrip/exophilin-8 to mobilise granules for fusion from the actin network to the plasma membrane after stimulation.2. Mlph and exophilin-8 interact via the exocyst3. Down-regulation of exocyst affects exocytosis in cells expressing exophilin-84. Exophilin-8 promotes fusion of granules docked by granuphilin at the membrane5. Exophilin-8 not required for Grph related granule docking at the plasma membraneA model for how the three effectors coordinate ISG secretion. According to this model there are 2 insulin secretion pathways in b-cells; (a) where Exo8 acts upstream of Mlph and with actin/Myosin Va/VIIa, exocyst and syntaxin 4 to move dock granules in actin and promote exocytosis, and (b) where Exo8 works in an antagonistic manner with Grph promoting secretion of granules docked at the membrane by Grph.In support of 1 they show that:insulin granules are docked at the cortex (a prelude to fusion) in WT and MlphKO, but not Exo8KO and double ME8DKO b-cells (Figure 1).In ME8DKO cells they show that expression of Exo8 but not Mlph restores cortical distribution of granules (Figure 1).Using TIRFM analysis of insulin secretion in pancreatic b-cells that fusion of non-resident granules (passenger and visitor types) is reduced in MlphKO cells compared with controls in response to stimulation (Figure 1).In support of 2 they show that:over-expressed/tagged and endogenous Exo8 and mlph can be co-precipitated in MIN6 cells (Figure 2).exocyst components Sec6 and Sec10 co-IP with Mlph and Exo8 (Figure 2).Depletion of sec10 reduces Mlph association with sec6 positive insulin granules (Figure 3).However, it would be important to show this directly e.g., by showing the results of Mlph/Exo8 IP in exocyst depleted cells to confirm that Mlph/Exo8 interaction is via the exocyst.In support of 3 they show that:using ELISA assays that glucose dependent insulin secretion is reduced in wild-type, but not Exo8KO, cells when Sec10 is KD (Figure 4A).In support of 4 they show that:insulin secretion in double granuphilin (Grph)/Exo8 DKO b-cells is reduced compared with single Grph KO cells (Figure 5A). Related to this they show that granule number in the TIRF field is affected GrphKO and GE8DKO but not Exo8KO. This seems surprising given that in Figure 1 the authors show that the number of docked granules is increased in Exo8KO cells.In support of 5they use TIRFM to show that granules are docked at the membrane by Exo8KO but not GrphKO or GE8DKO (Figure 5B).Overall, their data support the model shown in Figure 6.This is an interesting study but leaves open some important questions regarding insulin secretion and the role of Rab27a and its effectors in this process.The finding that GE8DKO islets release more insulin than wild-type (Figure 5A) is striking and suggests that another pathway(s) for insulin secretion exists in these preparations. Can the authors discuss the possible molecular mechanism of this pathway in their manuscript/include this in their model to present a more complete picture of the mechanisms of granule exocytosis?

It is possible that other Rab27 effectors compensate for the absence of Granuphilin and Exophilin-8. For example, exophilin-7 is involved in undocked granule exocytosis in the absence of granuphilin, although exophilin-7-knockout β cells display only modestly reduced exocytosis (Wang et al., Mol Biol Cell, 2013). However, we would like to suggest the possibility that secretory vesicles can fuse efficiently once they are directed to the plasma membrane using machineries such as cytoskeletons and/or the exocyst that are present in almost all cells. For example, in PC12 cells, SNAREs in native plasma membranes are constitutively active even if not engaged in fusion events (Lang et al., J Cell Biol 2002;158:751–760). Therefore, regulated exocytic pathways are thought to be evolved by acquiring additional gating machinery to prevent spontaneous or inappropriate fusion. We would also like to suggest that granule clustering sites in the actin cortex or along the plasma membrane reflect the presence of such gating steps (in contrast, no vesicles accumulate near the target membrane in constitutive exocytic pathways). We would like to propose that Rab27 effectors play such regulatory roles, rather than essential roles, in exocytosis, as discussed previously (Izumi, Cell Struct Funct, 2021). If so, the finding that GE8DKO cells release more insulin than WT cells is not so surprising. I briefly describe our thought in Discussion as below.

In summary, our findings are the first to determine the functional hierarchy among different Rab27 effectors expressed in the same cell, although the exact mechanism remains unknown how these different steps are regulated in a coordinated manner.

It should be noted, however, that granule exocytosis is not abrogated in β cells lacking any of the single or double Rab27 effectors. This may suggest the existence of other secretory pathways. For example, other Rab27 effectors, such as exophilin-7 (also known as JFC1 or Slp1), can also promote some types of exocytosis (Wang et al., 2013). However, given that granules can arrive close to the plasma membrane and fuse efficiently without prior capture in the actin cortex by exophilin-8 and/or stable docking to the plasma membrane by granuphilin, Rab27 effectors are thought to be evolved to play regulatory roles that prevent spontaneous or inappropriate fusion in regulated secretory pathways, as suggested previously (Izumi, 2021).

Related to this while I understand that it may be out of the remit of this project to characterise Grph/Exo8 independent pathways from scratch it would nevertheless be highly relevant to this study on Rab27a effectors for the authors to investigate the possibility that Rab27 and Mlph are involved in this pathway. For example, by generating triple Mlph, Exo8 and Grph8 KO islets or depleting in Mlph in GE8DKO islets and examining the effect of this on exocytosis and how this relates to exocytosis in the absence of Rab27a.

If melanophilin has function in the absence of exophilin-8, ME8DKO cells should show additional defects compared with Exo8KO cells, which we could not detect (Figure 2). Furthermore, considering that melanophilin deficiency only affects the passenger type exocytosis (Figure 2D), we think it unlikely that additional melanophilin deficiency suppresses the increase in the resident type exocytosis in GrphKO or GE8DKO cells (Figure 6). Furthermore, as described above, constitutive-like exocytic pathways or other effectors such as exophilin-7 may function in the absence of granuphilin and exophilin-8.

Regarding the possible different pathways have the authors noticed any differences in the kinetic profile of fusion events as monitored by TIRFM this might provide interesting insight into the relationship and mechanism between the proposed different pathways.

Although we have not examined differences in kinetic profiles of fusion events between WT and mutant cells, we previously showed no obvious differences in time courses or amplitudes of fluorescence intense changes during fusion reaction on average, among resident, visitor, and passenger types of exocytosis in WT cells (Kasai et al., Traffic 2008; Wang et al., Diabetes 2020).